

# Dew frequency across the US from a network of in situ radiometers

François Ritter[1], Max Berkelhammer[1], and Daniel Beysens[2]

[1]Department of Earth and Environmental Sciences, University of Illinois at Chicago, Illinois, USA
[2]Physique et Mécanique des Milieux Hétérogènes, CNRS, ESPCI, PSL Research University, Sorbonne Université, Sorbonne Paris Cité, Paris, France and OPUR, Paris, France

**Correspondence:** François Ritter (fritte2@uic.edu)

**Abstract.** Dew formation is a ubiquitous process but its importance to energy budgets or ecosystem health is difficult to constrain. This uncertainty arises largely because of a lack of continuous quantitative measurements on dew across ecosystems with varying climate states and surface characteristics. This study analyzes dew frequency from the National Ecological Observatory Network (NEON), which includes 11 grasslands and 19 forest sites from 2015 to 2017. Dew formation is determined at 30 min intervals using in situ radiometric surface temperatures from multiple heights within the canopy along with meteorological measurements. Dew frequency in the grasslands ranges from $15\%$ to $95\%$ of the nights with a strong linear dependency on the nighttime relative humidity (RH), while dew frequency in the forests is less frequent and more homogeneous ($25 \pm 14\%$, 1 Standard Deviation). Dew mostly forms at the top of the canopy for the grasslands due to more effective radiative cooling and within canopy for the forests because of higher within canopy RH. The high temporal resolution of our data showed that dew duration reaches maximum values ($\sim$ 6-15 h) for RH $\sim 96\%$ and for a wind speed of $\sim 0.5\,\mathrm{m\,s^{-1}}$, independent of the ecosystem type. While dew duration can be directly assessed from the observations, dew yield needs to be estimated based on the Monin-Obukov Similarity Theory. We find yields of $0.14 \pm 0.12\,\mathrm{mm\,night^{-1}}$ (1 SD from nine grasslands) similar to previous studies, and dew yield and duration are related by a quadratic relationship. The latent heat flux released by dew formation is estimated to be non-negligible ($\sim 10\,\mathrm{W\,m^{-2}}$), associated with a Bowen ratio of $\sim 3$. The radiometers used here provide canopy-averaged surface temperatures, which may underestimate dew frequency because of localized cold points in the canopy that fall below the dew point. A statistical model is used to test this effect and shows that dew frequency can increase by an additional $\sim 5\%$ for both ecosystems by considering a reasonable distribution around the mean canopy temperature. The mean dew duration is almost unaffected by this sensitivity analysis. In-situ radiometric surface temperatures provide a continuous, non-invasive and robust tool for studying dew frequency and duration on a fine temporal scale.

## 1 Introduction

Natural ecosystems are expected to experience more water stress due to a shift in the frequency, intensity and duration of droughts in the context of climate change (Dai, 2013). Second-order processes in the water cycle such as fog deposition (Dawson, 1998), water vapor adsorption (Kosmas et al., 1998) or dew formation (Monteith, 1957) may alleviate or exacerbate water-stress on ecosystems depending on how the frequency of these events change with rising temperatures. These processes can have a direct impact on the water balance of an ecosystem via the Foliar Water Uptake mechanism, which allows some



plant species to rehydrate themselves by absorbing the residual water on their leaves (Boucher et al., 1995; Munne-Bosch et al., 1999; Burgess and Dawson, 2004). In the example of a Redwood forest in California, $80\%$ of the dominant species have been proven to possess this trait and it contributed to an increase up to $11\%$ in their leaf water content (Limm et al., 2009). The impact on the water balance can also be indirect via transpiration suppression (Tolk et al., 1995), with a potential dew-induced decrease of $30\%$ in transpiration rate (estimated from isotopic methods, Gerlein-Safdi et al., 2018) or an impressive $100\%$ increase in the water use efficiency between the dew-affected and the no-dew populations in a semi-arid area (using a photo-synthesis and transpiration rate monitor, Ben-Asher et al., 2010). The energy budget of the ecosystem will be affected as well, not only because of a change in the sensible and latent heat fluxes, but also in the emissivity of the surface. An increase of $10\%$ in the surface albedo due to the presence of dew has been reported from an analysis using satellite data (Minnis et al., 1997). All these combined effects on the water and energy balance might enable animal and plant species to survive longer droughts in the future, see Wang et al. (2017b) for a review on the ecological significance of non-rainfall water inputs.

Dew formation is the water vapor condensation on a surface when the surface temperature drops below the dew point temperature of the air. It occurs almost exclusively at night when reduced input of short wave radiation generates a negative net radiation for surfaces. Based on a recent review (Tomaszkiewicz et al., 2015) that includes 25 studies using dew condensers, a typical dew frequency is $41\pm14\%$ of the nights (1 Standard Deviation or SD) and a typical dew yield is $0.13\pm0.10\,\mathrm{mm\,night^{-1}}$ (1 SD) on an average study period of one year. Dew formation is known to be abundant even in semi-arid areas that are in close proximity to a moisture source. For example, dew occurred during $78\%$ of the nights during a 4 year period in a semi-arid coastal steppe (Uclés et al., 2014) and $77\%$ of the nights during the growing season at the edge of a desert oasis (Zhuang and Zhao, 2017). The three most important limitations of dew formation are relative humidity, the efficiency of radiative cooling (driven by cloud cover, biomass density, vapor pressure of air and surface emissivity) and surface wind speed (influenced by regional climate, local topography and canopy structure). The relative humidity is directly related to the dew point temperature of the air, while the radiative cooling creates a difference in temperature between the surface of the canopy and the air at night. Lastly, the wind speed increases the heat exchange between the cold surface and the air, thus lowers the available energy for condensation and eventually reduces the dew yield. It is also important to note that the two extreme cases where the relative humidity is at $100\%$ or the wind speed is close to $0\,\mathrm{m\,s^{-1}}$ are not beneficial for dew formation either. The radiative cooling is lowered or even canceled in the first case ("radiative fog") and in the second case, some convection is needed to replenish the available water vapor in the boundary layer of the canopy. Dew formation is therefore a subtle process that requires an adequate balance between relative humidity, radiative cooling and wind speed.

So far, no unique standardized method currently exists to measure dew formation. The previously mentioned radiative pas-sive condenser (RDC) is usually a flat tilted plate (area of $\sim 1\,\mathrm{m^2}$), thermally isolated from the ground, and from which dew droplets are collected under gravity flow (Maestre-Valero et al., 2015). It is also possible to weigh materials (tissues, woods, plastic) before and after the dew event to calculate the dew yield (Pan et al., 2010), or to use leaf wetness sensors to estimate the presence/absence of wetness on its surface (Moro et al., 2009; Berkelhammer et al., 2013). These approaches provide a po-



tential dew frequency and yield but the artificial surfaces used to collect dew droplets are not representative of the ecosystem in terms of thermal and radiative properties like the conductivity or emissivity (Sentelhas et al., 2004; Maestre-Valero et al., 2011).

Additionally, natural ecosystems are dynamic environments that vary seasonally and grow or decline over decades. Long-term dew formation measurements therefore need to capture this evolution in the canopy. For example, an approach that is more ecosystem-focused takes advantage of large lysimeters to weigh dew formation at night on natural ecosystems (Meissner et al., 2007). In a study using two high precision lysimeters on temperate grasslands for 2 years, Xiao et al. (2009) shows that the maximum nighttime dew yield is limited by the growth of the canopy height for a maize plantation, from $0.1\,\mathrm{mm\,night}^{-1}$

(canopy height below $0.2\,\mathrm{m}$) to $0.5\,\mathrm{mm\,night}^{-1}$ (same site, but canopy height at $2\,\mathrm{m}$). This result is obviously influenced by the meteorological conditions associated with the growing season, but it also shows the importance of the canopy structure on the dew frequency and yield. To summarize, although extensive work has been done to measure dew formation there remains a lack of data using continuous measurements performed in different ecosystems with the same standardization. The measurement approach either uses a surface that is not representative of the properties and dynamic of the canopy (RDC, leaf wetness

sensors...), or generates a highly local measurement (lysimeters). It is therefore difficult to establish a global or continental network of standardized dew measurements over a sufficiently long period of time to analyze the impact of climate change on dew frequency, duration and yield. Such data are needed to validate dew models, which provide a tool to test the sensitivity of dew formation to different forcing mechanisms.

Models of dew formation are usually designed from an energy balance, like the Penman-Monteith equation (Monteith et al., 1965), where the latent heat released by condensation is the parameter to solve for. Otherwise, semi-empirical models are also reliable and are calculated from the relationships between dew formation and its drivers: relative humidity, wind speed and cloud cover (Beysens, 2016). These models are fed with classical meteorological measurements and they have been reasonably well validated with direct dew measurements. For example, in Beysens (2016), the average slope between the cumulated

experimental yield and the cumulated predicted yield is $0.8 \pm 0.2$ (1 SD) based on 10 studies using RDCs. Models give the opportunity to study dew formation on a larger scale, but only two recent attempts have been published: a global dew yield estimation based on reanalysis data (Vuollekoski et al., 2015) and a map of the dew yield on the Mediterranean coast based on data from a network of traditional weather stations (Tomaszkiewicz et al., 2016). These studies estimate dew yield using semi-empirical models, which consider the surface receiving dew formation as a flat plate to simplify the equations of heat

and vapor transfer. Both methods detect a significant change in the dew yield over a climatic time scale, for example $\pm 10\,\%$ change from 1979 to 2012 in different parts of the world (Vuollekoski et al., 2015) or a $27\,\%$ decrease in future summers in the Mediterranean basin from 2013 to 2080 (Tomaszkiewicz et al., 2016). However, these attempts did not have access to the surface temperature of the canopy and had to implicitly retrieve it from an energy balance or assume it close to the dew point temperature. It is well known in the Eddy-covariance field that the energy balance closure is less reliable at night simply

because of the higher uncertainties in the latent and sensible fluxes associated with a decrease in temperature and wind speed after sunset (de Roode et al., 2010). Indeed, as the atmospheric stability increases at night, the turbulent component of the energy transfer decreases relative to the radiative component. The surface temperature retrieval is therefore associated with a





high uncertainty, as it depends on the net radiation, the ground flux (driven by the soil temperature gradient), the wind speed, the air temperature and the exchange of latent heat from condensation or possible nocturnal evapotranspiration. This problem can be bypassed by estimating the surface temperature using in-situ radiometric surface temperature measurements of different canopies, and then simply compare them with the dew point temperatures of the air.

The scientific literature about infrared thermal radiometers (or IR-radiometers) is abundant in the remote sensing field (see Kalma et al. (2008) for a review), because satellites offer the opportunity to estimate the sea or land surface temperature on a global scale. However, in-situ radiometric measurements of the surface temperature in natural ecosystems is less well-documented (Stewart et al., 1994; Troufleau et al., 1997; Suleiman and Crago, 2004; Payero and Irmak, 2006; Sánchez et al., 2008, 2011, 2014). Studies of the nighttime surface temperatures using this approach are even more limited due to the overall lack of interest in the nocturnal processes when the latent/sensible heat is limited and there is no photosynthetic activity. The surface temperature is retrieved from the infrared emissions using the Stephan-Boltzmann law, and then is usually corrected with respect to the surface emissivity. For the new generation of IR-radiometers (Apogee SI-111, used in this study), the order of magnitude of the uncertainty is $\pm 0.5\,°C$ (Payero and Irmak, 2006) associated with a response time of $0.6\,s$. Two studies have observed that the dew point temperature of the air is above the in-situ radiometric surface temperature when dew droplets are collected on a RDC (Maestre-Valero et al., 2015) or weighted by a lysimeter (Jacobs et al., 2006). The IR-radiometers are therefore suitable for studying dew frequency and dew duration. However, the dew yield is a parameter that is more difficult to constrain because it requires an estimate of the aerodynamic resistance between the leaf surface and the air for vapor transfer, which depends on the wind speed, canopy structure and atmospheric stability.

IR radiometers are a standard component of the National Ecological Observatory Network (NEON) tower network, which was established in 2012 to assess the change of ecosystems on a continental-scale over 30 years. NEON recently released preliminary data that includes traditional meteorological measurements and radiometric temperature of the canopy of 11 grasslands (2748 nights) and 19 forests (4747 nights) in the USA with a 30 min resolution from 2015 to 2017 (Fig. 1). This network has the advantage of providing a measurement that is continuous, captures an average condition for the whole canopy of diverse ecosystems, does not rely on artificial surfaces nor does it require solving the energy balance for canopy or soil surfaces at night. We focus in this paper on the growing season when temperatures are above $0\,°C$ (frost events are therefore excluded). We analyze the vertical variability of the relative humidity and surface temperature associated with each site at night (Fig. 2); show how the nighttime radiative cooling is limited by the relative humidity and drives the difference of temperature between the air and the surface (Fig. 3); present the dew frequency, duration and yield in forest and grassland ecosystems across very different climatic conditions (Fig. 4 and Fig. 5); estimate and compare the sensible, latent and radiative flux associated with a dew event (Fig. 6); constrain the sensitivity of dew duration to relative humidity (Fig. 7) and wind speed (Fig. 8); describe the nocturnal cycles associated with the absence or presence of dew formation (Fig. 9 and Fig. 10) and finally interrogate the possible importance of the spatial variability of the surface temperature on the dew frequency using a Monte-Carlo sensitivity



analysis (Fig. 11).

## 2  Method

*Description of the data*. All the analyses performed in this study are based on the preliminary dataset recently released by
NEON with a 30-min resolution from 2015 to 2017 (downloaded from http://data.neonscience.org on 10 April 2018). Meteo-
rological parameters were measured at different heights from a tower (Fig. 1), and we focused on 11 grasslands (2748 nights)
and 19 forests (4747 nights) with the following measurements available from 16:00 to 10:00 next day (local time used in this
paper): radiometric surface temperature ($T_s$, Apogee SI-111), relative humidity (RH, Vaisala HMP155), pressure (P, Vaisala
PTB330), air temperature ($T_a$, Met One Instruments 62789 Aspirated Radiation Shield), wind speed (WS, Gill's WindObserver
II 2D Sonic Anemometer) and net radiation above canopy ($R_n$, NR01 Net Radiation Sensor). The code name, sample size, ele-
vation, location and canopy height of each site are reported in Table 1 and Table 2. The data were separated into two layers of
vegetation: "within canopy" (spatial average of the wind speed, air and surface temperature measurements below the top of the
canopy) and "at the top of the canopy" (using the IR-radiometer, air and wind speed sensors closest to the top of the canopy).
For the grasslands, the RH sensor close to the ground is used for both layers, because the other sensor is too far away from the
top of the canopy ($\sim 8\,\mathrm{m}$, see Fig. 1). For the forests, the RH sensor close to the ground is used for the layer "within canopy"
and the RH sensor above canopy ($\sim 15\,\mathrm{m}$) is used for the layer "at the top of the canopy". Although vertical interpolation could
have been used to estimate RH co-located with the canopy temperature measurement, this would have been inappropriate here
because the canopy structure or atmospheric stability does affect the vertical profile of RH. Finally, a dew point temperature
($T_d$) is calculated for both layers following the equation in Wagner and Pruß (2002) and using the respective $T_a$ and RH of each
layer.

*IR-radiometers information*. The IR-radiometer used by NEON is the Apogee SI-111, with a response time of 0.6 s and
an associated uncertainty of $\pm 0.5\,^{\circ}\mathrm{C}$. The spectral range is 8 to 14 μm (atmospheric window) and the IR-radiometer is fully
operational at RH $= 100\,\%$. NEON applies a correction for temperature of the sensor, however the radiometric measurements
are not corrected with respect to the emissivity of the surface (considered here as 1), because of insufficient information on
the plant type, vegetation density or soil type. By summing the individual uncertainties in quadrature, NEON estimates the
combined uncertainty of each measurement as $\pm 0.6\,^{\circ}\mathrm{C}$ for 30-min averages during nighttime. This uncertainty will be consid-
erably reduced by the calculation of mean values on a nightly time scale and then over the full period of study ($\sim 250$ nights
per site on average) so that the statistic of our study can be considered as robust thanks to the sample size.

*Data filtering*. Only the nights during the growing seasons have been selected (mean surface and air temperature above
$0\,^{\circ}\mathrm{C}$ each 30-min steps) so that frost events can be excluded from the analysis. Nights with more than $25\,\%$ of missing data
(or negative temperatures, or obvious outliers) from 16:00 to 10:00 next day have been removed from our dataset. Obvious





outliers are flagged by calculating the absolute difference between the value of a sensor and the median value of all the sensors measuring the same parameter (limit of $30\,\mathrm{m\,s^{-1}}$ for the wind speed and $40\,^{\circ}\mathrm{C}$ for the air and surface temperature). Rainy

nights are included in the dataset.

*Definition of a dew event, dew duration and dew frequency.* A 30-min dew event occurs when $\mathrm{T_s} \leq \mathrm{T_d}$ (time window from 16:00 to 10:00 next day), the dew duration per night is the summation of 30-min dew events during the night, the dew duration per site is the mean value of the strictly positive dew durations per night and the dew frequency is the percentage of the number

of dewy nights (nights with at least one dew event of 30-min) compared to the total number of nights (Table. 1 and Table 2). In Fig. 6-8, the meteorological parameters are averaged during the dew event (i.e., over the dew duration), while in Fig. 2-4 they are averaged from 23:00 to 5:00 next day.

*Calculation of the dew yield, latent heat flux, sensible heat flux for the grasslands.* To estimate the dew yield, latent and

sensible heat flux, we must calculate the aerodynamic resistance to vapor transport $r_{av}$ between the surface and the air, which depends on the wind speed, canopy height and atmospheric stability in the Monin-Obukhov Similarity Theory (MOST). Our work precisely follows Jacobs et al. (2006) for the calculation of $r_{av}$ based on the computation of the bulk Richardson number (BRN), which is an indicator of the atmospheric stability and defined as the ratio of buoyancy production divided by the shear production. Empirical flux-gradient relationships between the BRN and stability parameters are then used to compute

the integrated stability functions for momentum and vapor, which lead to the calculation of $r_{av}$ (see Jacobs et al., 2006). The latent heat flux (LE, in $\mathrm{W\,m^{-2}}$), sensible heat flux (H, in $\mathrm{W\,m^{-2}}$) and dew yield (M, in $\mathrm{mm}$) are then computed on a 30-min time scale following:

$$\mathrm{LE} = \mathrm{L} \times \rho_\mathrm{a} \times \frac{q(\mathrm{T_a}) - q^{\mathrm{sat}}(\mathrm{T_s})}{r_{av}} \tag{1}$$

$$\mathrm{H} = \mathrm{C_p} \times \rho_\mathrm{a} \times \frac{\theta_\mathrm{a} - \theta_\mathrm{s}}{r_{av}} \tag{2}$$

$$\mathrm{M} = \mathrm{E} \times \frac{\tau}{\rho_\mathrm{w}} \tag{3}$$

With $\rho_\mathrm{a}$ and $\rho_\mathrm{w}$ as the density of air and water ($\mathrm{kg\,m^{-3}}$), L the latent heat of condensation ($\mathrm{m^2\,s^{-2}}$), $\theta$ the potential temperature (K), $q$ the specific humidity (with sat standing for saturated, unitless), $\mathrm{C_p}$ the specific heat capacity ($\mathrm{m^2\,s^{-2}\,K^{-1}}$), $r_{av}$ the aerodynamic resistance ($\mathrm{s\,m^{-1}}$) for heat or vapor transport (commonly assumed to be similar) and $\tau$ the length of a time step (here $30 \times 60 \times 1000\,\mathrm{s}$, with the factor 1000 to convert M into $\mathrm{mm}$). The sign convention for H and LE (and any flux of

energy) is positive when the grasslands receive energy. So, during a dew event, H>0 and LE>0 per definition. It is important to note that the work of Jacobs et al. (2006) does only concern grasslands because the aerodynamic resistance is not well-defined within the canopy of the forests.



*Nocturnal cycles (Fig. 9 and Fig. 10)*. The analysis on a nocturnal scale is performed on 5 different populations. Three populations of grasslands based on their nighttime relative humidity: "dry" (RH $< 80\,\%$, CPER, JORN, KONZ, MOAB, OAES, ONAQ), "temperate" ($80\,\% <$ RH $< 90\,\%$, DCFS, HEAL, WOOD) and "tropical" (RH $> 90\,\%$, DSNY, LAJA). Two popula-
tions of forests based on the timing of dew formation: "early" ($\sim$21:00, DELA, SERC, STEI, BART, LENO, JERC, HARV, TREE") and "late" ($\sim$00:00, DEJU, OSBS, MLBS, CLBJ, TALL, UNDE). Each of these five populations has been cut into two sub-ensembles: nights with the presence of at least 30-min of dew formation (left column in Fig. 9 and Fig. 10), and nights with a complete absence of dew formation (right column). For each night and for each parameter (wind, $T_s$, $T_a$, $T_d$), the mean value from 16:00 to 10:00 next day has been subtracted to produce nocturnal anomalies. These nocturnal anomalies have been
stacked and the standard deviation has been calculated for each half-hour. Finally, we add the mean value of the previously subtracted mean values to the stack to produce the final nocturnal cycles.

*Sensitivity analysis of the spatial variability of $T_s$ on dew formation (Fig. 11)*. To simplify the analysis, the spatial distribution of the surface temperatures around the mean value measured by the radiometer on a 30-min time scale is assumed to
be Normal. We have tested the sensitivity of dew frequency and dew duration to a change in the standard deviation ($\sigma$) of the Normal distribution, from 0 to 0.5. For each half hour and for a given $\sigma$, we use a Monte-Carlo simulation (or MCS, n $=$ 1000 runs) following a Normal distribution around the 30-min average of $T_s$, and we calculate the percentage of randomized points (X$\,\%$) that fall below $T_d$ among these 1000 runs. In this context, X is defined here as the percentage of the canopy receiving dew formation (see Fig. 11, panel a). If X $< 5\,\%$, we consider that no dew event is occurring for this half-hour. We finally calculate
for a given $\sigma$ the dew frequency, duration and the averaged percentage of canopy receiving dew formation (mean value of all percentages above $5\,\%$) for the grasslands and the forests. If $\sigma = 0$, the percentage of the canopy receiving dew formation is either $0\,\%$ (absence of dew) or $100\,\%$ (presence of dew) and the results are equivalent to the analysis without MCS. When $\sigma > 0$, more dew will be formed during the night, but on a lower percentage of the canopy.

# 3   Results and discussion

## 3.1   Analysis for each site

### 3.1.1   Relative humidity, radiative cooling and temperature

The 11 grasslands and 19 forests can be characterized at night by their mean surface temperature and mean relative humidity (from 23:00 to 5:00 next day, local time) over the growing season. A general positive trend is observed in Fig. 2 (panel a) and is
mostly explained by the effect of relative humidity on radiative cooling (Fig. 3, panel a). Indeed, a higher water vapor content will make the air more efficient at absorbing and re-emitting longwave radiation, therefore the nighttime boundary layer will be more opaque to radiative loss and this mechanism will keep the surface temperature warm at night. Additionally, sites with a high elevation like MOAB ($1779\,\mathrm{m}$), ONAQ ($1662\,\mathrm{m}$) and RMNP ($2742\,\mathrm{m}$) are associated with cold and dry air due to the



low pressure. The relative humidity within the canopy of the forests is usually higher than the grasslands at night ($86 \pm 9\%$ vs $76 \pm 15\%$, 1 SD) not only because of a denser canopy structure, but also because forests occur in wetter regions than grasslands (Woodward et al., 2004). The two exceptions here being the grasslands LAJA and DSNY, which present tropical characteristics

(RH $= 95 \pm 3\%$, 1 SD).

     Measurements performed at different heights provide the opportunity to study the vertical variability in the surface temperature and relative humidity. The difference of surface temperature between the bottom and the top of the canopy (Fig. 2, panel b) is similar at night for the forests ($0.3 \pm 0.5\,^{\circ}C$, 1 SD) and the grasslands ($0.7 \pm 0.6\,^{\circ}C$, 1 SD) based on a t-test ($p = 0.07 > 0.05$).

One could consider that the vertical variability in the surface temperature at night is larger for tall-canopy ecosystems than low-canopy ecosystems because radiative fluxes are exchanged in a bigger volume. Although forests have a much higher canopy height than the grasslands (5-40 m of difference, Fig. 1), the vertical variability in their surface temperature is not larger at night. One possible explanation is that forests are wetter and have more biomass than grasslands. Their canopy will therefore cool down slower as will the ground because of a higher heat capacity and inefficient radiative cooling (Fig. 3, panel a). The

positive relationship observed among the forested sites in Fig. 3 (panel b) seems not be primary driven by the canopy height, and the ground flux or cloud cover data (absent from this analysis) might play an important role in the vertical gradient of surface temperature at night, as well as the vegetation density and the site wetness. Looking now at the relative humidity, it reveals to be always higher within the canopy than above canopy for both ecosystem types (Fig. 2, panel c). Indeed, air parcels within the canopy are not effectively mixed with drier air higher in the atmosphere (low wind speed within canopy) and the

combination of soil evaporation and transpiration provides continual moisture to the canopy air space.

     The difference in temperature between the air and surface ($T_a - T_s$) is mainly driven by radiative cooling for most sites (Figure 3, panel c). Interestingly, three grasslands (LAJA, DSNY and HEAL) and one forest (DEJU) with the shortest canopy height (6 m) show a large $T_a - T_s$ despite minimal radiative cooling (Fig. 3, panel c). These four sites do not belong to the

same population: HEAL and DEJU are both located in Alaska (cold and dry air) while LAJA and DSNY are tropical (warm and wet air). Moreover, HEAL and DEJU are the only two sites that are not part of the trend in the relationship between $R_n$ and RH (Fig. 3, panel a). It is not clear at this point what drives these sites away from the otherwise strong relationship. The ground flux could act as a sink of energy at night for these four sites, and could decrease the temperature of the canopy, which would consequently reduce the radiative cooling. Another explanation could be that nocturnal atmospheric dynamics such as

the presence of warm convection above canopy (increasing the difference of temperature) associated with an opaque cloud cover (drastically reducing the radiative cooling, even if the relative humidity is below $85\%$ for HEAL and DEJU, see Fig. 3, panel a). Our current data are not sufficient to verify these hypotheses and further investigations are required to understand what is driving $T_a - T_s$ when the radiative cooling is weak. In a study using RDCs in tropical areas (Tahiti and Tikehau, French Polynesia), Clus et al. (2009) does observe that the relative humidity impacts the efficiency of the radiative cooling, however

these sites cannot be compared to LAJA or DSNY because of the presence of strong winds, a much lower relative humidity



($\sim 80\,\%$ at night compared to $\sim 95\,\%$) and the absence of surface temperature data except for one night.

### 3.1.2 Dew frequency, duration and yield across sites

The most frequent location where dew forms is on the top of the canopy of grasslands, which is much cooler than the ground
(Fig 2, panel b) and within the canopy of the forests, where the relative humidity is much higher than above canopy (Fig 2, panel c). Indeed, the dew frequency within the canopy of the grasslands is only $7\,\%$ compared to $29\,\%$ at the top of the canopy (DSNY and LAJA excluded because of their tropical characteristics). For forested sites, dew forms only $11\,\%$ of the nights at the top of the canopy of the forests compared to $25\,\%$ within canopy, although the position of the RH sensors above canopy for the forests is far away from the top of the canopy (difference of $\sim 15\,\mathrm{m}$) and that is why dew frequency at the top of the canopy
might be underestimated for the forested sites. This study will therefore focus on the dew formation at the top of the canopy of the grasslands and within the canopy of the forests only.

The grassland sites show a large gap in temperature between the air and the surface at night (Fig. 3, panel c). Consequently, the main limitation of dew formation at the top of the canopy of the grasslands will be the relative humidity. A threshold of
$\mathrm{RH} = 75\,\%$ separates two patterns for the dew frequency in the grasslands (Fig 4, panel a): (i) a low and constant dew frequency of $\sim 15\,\%$ of nights for $\mathrm{RH} < 75\,\%$ and (ii) a linear increase in dew frequency with rising relative humidity (slope of 3.5, $r^2 = 0.92$) for $\mathrm{RH} > 75\,\%$. We have isolated three populations of grasslands based on their relative humidity: "dry" (six sites with $\mathrm{RH} < 80\,\%$), "temperate" (three sites with $80\,\% < \mathrm{RH} < 90\,\%$) and "tropical" (two sites with $\mathrm{RH} > 90\,\%$). The dry, temperate and tropical sites have a dew frequency of $15 \pm 6\,\%$ (1 SD), $59 \pm 4\,\%$ (1 SD) and $95 \pm 4\,\%$ (1 SD), respectively. This
dew frequency versus relative humidity relationship revealed by several grasslands with different climatic conditions has not been yet reported in the literature for the following reasons: the former dew studies were limited to one or two sites, there is a lack of standardized measurements between them, there is an absence of continuous humidity measurements or a decision to not report them (Xiao et al., 2009), and finally the sites are mostly chosen for their high nocturnal RH, especially in arid areas close to a coast (Zangvil, 1996; Muselli et al., 2009; Moro et al., 2009; Uclés et al., 2014; Maestre-Valero et al., 2015).
The in-situ radiometric network from NEON has not been specifically designed to work in locations with high dew yield, and thus the data provide information on sites that span a continuum from exceptionally low dew frequencies to almost nightly. However, former dew studies focusing on a single site have reported a similar RH threshold values below which dew has a low probability to form: $\mathrm{RH} = 78\,\%$ in Wang et al. (2017c) (204 nights in a semi-arid site in China using leaf wetness sensors), and $\mathrm{RH} = 75\,\%$ in Guo et al. (2016) (166 nights in a cold desert in China using eddy-covariance data). Even if the linear relation-
ship in Fig. 4 (panel a) is strong, the relative humidity alone is not sufficient to predict dew frequency with a good accuracy (the residuals have a standard deviation of $10\,\%$), which highlights the role of canopy structure and wind speed in dew formation (Sect. 3.2). See Beysens (2018) for a detailed explanation of the relationship between dew formation and relative humidity.





As mentioned in the previous sections, forested sites have a higher relative humidity within canopy than the dry and temperate grasslands at night (Fig. 2, panel a) but a lower air/surface temperature gap (Fig. 3, panel c). The relationship between dew frequency and relative humidity within the canopy can be still considered as linear for RH $> 80\,\%$ (Fig 4, panel a, slope of 2.3), but with a much lower coefficient of determination ($r^2 = 0.31$) because the inefficient radiative cooling or very low wind speed within canopy are also limiting dew formation in these ecosystems. Forests have a mean dew frequency of $25 \pm 14\,\%$ (1 SD), which is between the dry and temperate grasslands. Interestingly, the forest DEJU is consistent with the linear relationship calculated from the population of grasslands (Fig. 4, panel a) certainly because it has the shortest canopy height ($6\,\mathrm{m}$) among the forested sites. Dew formation is less well-documented in forested sites than arid areas or croplands/grasslands, possibly because small inputs from dew are not likely to have a significant impact on these ecosystems, or because their tall canopies are considered as a natural radiative barrier to dew formation. However, Hao et al. (2012) has reported a dew frequency of $73\,\%$ over 142 nights using Eddy-covariance data in a hyper arid forest (canopy height 8-10 m, forest coverage of $50\,\%$, annual rainfall of $35\,\mathrm{mm}$). The much higher dew frequency might be explained by the short canopy height, the low vegetation density, the presence of a river close to the area of study or simply the poor performances of the Eddy-covariance method at night (de Roode et al., 2010). Unfortunately, nighttime relative humidity measurements are not available and cannot be compared with our study. In a sparse Elm wood site (unknown canopy height, forest coverage of $40\,\%$, annual rainfall of $385\,\mathrm{mm}$), Wang et al. (2017a) has measured a dew frequency of $54\,\%$ over 80 nights using weighing methods, which is much more consistent with our data. Again, relative humidity measurements are not available for the period of study.

We define dew duration as the time during which the dew point temperature is above the surface temperature at night. The dry and temperate grasslands have a dew duration of $3.4 \pm 1.1\,\mathrm{h}$ (1 SD), and the forests $4.5 \pm 1.1\,\mathrm{h}$ (1 SD). A comparison with former studies is difficult because of the varying range of dew duration in the literature: from 1-4 h (Pan et al., 2010; Hao et al., 2012) to 4-6 h (Wang et al., 2017a) or above 6 h (Moro et al., 2009; Uclés et al., 2014; Guo et al., 2016) depending on the climatic conditions, canopy structure and measurement protocol. Dew duration is positively affected by the relative humidity (Fig 4, panel b), but it is also sensitive to a change in windspeed (see Sect. 3.2 for a detailed analysis of the sensitivity of dew duration to these parameters). The two tropical grasslands (LAJA and DSNY) show a much higher mean dew duration ($8.8 \pm 0.1\,\mathrm{h}$, 1 SD) compared to the rest of the sites (Fig 4, panel b). Both have high nocturnal relative humidity (RH $= 95 \pm 3\,\%$, 1 SD) with a surprisingly high difference of temperature between the air and the surface of the canopy at night ($\mathrm{T_a} - \mathrm{T_s} = 2.1 \pm 0.5\,^\circ\mathrm{C}$, 1 SD) despite having the most inefficient radiative cooling due to the saturated air (see Fig. 3, panel c). As previously mentioned, this result needs further investigation to be fully understood, but it does not look unrealistic as former studies have reported longer dew durations, for example $9.6 \pm 3.2\,\mathrm{h}$ (1 SD) in a coastal steppe over a 4 years period (Uclés et al., 2014).

So far, two robust results (dew frequency and dew duration) have been presented and calculated based on the sign of $\mathrm{T_d} - \mathrm{T_s}$, without needing to consider atmospheric stability, which is essential to estimate the dew yield (expressed in $\mathrm{mm\,night^{-1}}$). The dew yield estimation for the grasslands is based on the calculation of the aerodynamic resistance to vapor transport, which depends on many variables and is therefore prone to more uncertainties. Figure 5 (panel a) shows the dew yield versus dew du-



ration relationship using the nightly average data of the grasslands, and only one site (DSNY) failed to produce a realistic dew yield (usually below $1\,\mathrm{mm\,night^{-1}}$). DSNY is a tropical site with frequent rain and fog events that might largely contribute to its nocturnal water balance and reduce the performance of the MOST, even if the dew duration and frequency of DSNY are consistent with the other tropical site (LAJA). A quadratic curve ($y = a \times x^2$, $a = 0.0056\,\mathrm{mm\,hours^{-2}}$) fits remarkably well ($r^2 = 0.94$) the dew yield versus dew duration for the other grassland sites (Fig. 5, panel b). The fit has been calculated on 22 median values (30 min resolution on a dew duration below $11\,\mathrm{h}$), and the mean value of the residuals (nightly data) is $0.02 \pm 0.16\,\mathrm{mm\,night^{-1}}$ (1 SD, 575 nights). A similar quadratic pattern can be observed in the dew yield versus dew duration relationship for two croplands in (Meng and Wen, 2016) using a flux-profile method and the MOST, but the author did not perform a regression analysis for comparison. The quadratic nature of the relationship can be qualitatively explained by the fact that the vapor pressure deficit ($q(\mathrm{T_a}) - q^{\mathrm{sat}}(\mathrm{T_s})$ in Eq. (1)) is positively related to the dew duration. This can be seen in Fig. 7 (panel b) knowing that $q(\mathrm{T_a}) - q^{\mathrm{sat}}(\mathrm{T_s}) = q^{\mathrm{sat}}(\mathrm{T_d}) - q^{\mathrm{sat}}(\mathrm{T_s})$. Per definition, the dew yield is the integral of the dew flux over the night, or more visually the area below the dew flux curve. This area is roughly equal to the dew duration multiplied by the mean dew flux, which is driven by $q(\mathrm{T_a}) - q^{\mathrm{sat}}(\mathrm{T_s})$ and depends also on the dew duration as explained above. For example, if the dew duration is divided by 2, the mean dew flux will approximatively be divided by 2 as well and the area will therefore be divided by 4, which explains the quadratic nature of the relationship.

The mean dew yield of the dry and temperate grasslands is $0.14 \pm 0.12\,\mathrm{mm\,night^{-1}}$ (1 SD), which is consistent with the 25 studies collecting dew on RDCs ($0.13 \pm 0.10\,\mathrm{mm\,night^{-1}}$, 1 SD, see Tomaszkiewicz et al. (2015)) and this provides strong support that the MOST produces an accurate estimation of the dew yield based on radiometric surface temperatures. The mean dew yield of the tropical site LAJA is $0.52\,\mathrm{mm\,night^{-1}}$, which is high, but frequent fogs might be included in this budget because of the exceptionally high vapor pressure deficit at LAJA (saturated air and large difference of temperature between the surface and the air). In a study based on weighing methods, Pan et al. (2010) has shown a significant change between dew yields forming in foggy conditions ($\sim 0.3\,\mathrm{mm\,night^{-1}}$) and non-foggy conditions ($< 0.1\,\mathrm{mm\,night^{-1}}$) that supports our result. Additionally, the maximum dew yield measured by high precision lysimeters in two temperate grasslands reached 0.5-$0.6\,\mathrm{mm\,night^{-1}}$ (Xiao et al., 2009) for 6 nights. These events were associated with meteorological conditions that might occur frequently in LAJA.

### 3.1.3 Estimation of three energy components during dew formation in grasslands

The importance of dew formation in the nocturnal energy budget is poorly documented because of the technical challenge in measuring fluxes from Eddy covariance during periods of low turbulence. Opinions differ, from dew being a negligible component of the energy budget that cannot be distinguished from the noise (de Roode et al., 2010) to dew as a process that significantly participates in the energy budget and helps to better constrain the energy closure (Jacobs et al., 2008). In this section, we compare the value of the sensible heat flux (H, $\mathrm{W\,m^{-2}}$), Latent heat flux (LE, $\mathrm{W\,m^{-2}}$) and net radiation above canopy (R$_n$, $\mathrm{W\,m^{-2}}$) associated with a dew event. See the Method Section to understand how H and LE have been calculated



with the MOST.

Our system is a grassland (biomass above ground, roots excluded), and the energy budget is described as a volume energy balance:

$$S = R_n + H + LE + G \qquad (4)$$

with S the energy lost by the system (storage term, $\mathrm{W\,m^{-2}}$), G the ground flux ($\mathrm{W\,m^{-2}}$) and here the term of advection has been neglected. The volume energy balance is calculated only during a dew event and the sign convention is to consider as positive a component bringing energy to the system. In this case, H and LE are always positive because a dew event is associated with the condensation of water vapor on a surface (LE>0), which can only happen when the air is warmer than the surface (H>0). Figure 6 shows the median value (error bars: 25 % and 75 % quantiles) of each energy budget component for each site (number of dewy nights indicated in parenthesis), the residuals H+LE+$R_n$ being equivalent to S-G.

The uncertainties associated with the MOST are high at night and sensitive to the gradient of temperature between the air and the leaf, but also the wind speed and stability of the atmosphere. For this reason, the discussion here will be only based on the order of magnitude of the components, and DSNY will be excluded from the analysis (see Fig. 5, panel a). In the following, we use the $50\%\{25\%,75\%\}$ quantiles notation. A typical latent heat flux released by dew formation is $11\{6,20\}\,\mathrm{W\,m^{-2}}$ for the grasslands (excluding LAJA and DCFS). The sensible heat flux for the same eight sites during dew formation is three times larger than LE and associated with a higher variability (H = $32\{18,55\}\,\mathrm{W\,m^{-2}}$), while the net radiation is twice as important as the sensible heat flux ($R_n = -64\{-71,-54\}\,\mathrm{W\,m^{-2}}$). For the tropical site LAJA, the latent heat flux can reach $36\{28,43\}\,\mathrm{W\,m^{-2}}$ and stays above the sensible heat flux (H = $23\{18,29\}\,\mathrm{W\,m^{-2}}$). The Bowen ratio (defined as H/LE) might therefore go below 1 during dew events in exceptionally wet sites (nighttime RH > 90 %).

As we will see in Sect. 3.3, the leaf temperature is rarely increasing during dew formation, despite the latent heat and sensible heat fluxes bringing energy to the system. The continuous decrease in leaf temperature means that the storage term S is negative during the night. The residuals (S-G) shown in Fig. 6 are positive for three sites: OAES, DCFS and LAJA. This result suggests one of the following: (i) The ground flux G is an important sink of energy (at least $\sim -30\,\mathrm{W\,m^{-2}}$) for these sites at night and/or (ii) H and LE are overestimated and the MOST fails to partition the energy budget for these sites at night. This study will not be able to validate or invalidate these two hypotheses. However, we can confirm the preliminary work of Jacobs et al. (2008) and affirm that the latent heat flux released by dew formation is a non-negligible component in the nocturnal energy budget, with a typical Bowen ratio of $\sim 3$ during a dew event, that is similar to Meng and Wen (2016) for two croplands using a flux-profile method (69 and 128 dewy nights, Bowen ratio of 1-3 in presence of dew).



## 3.2 Sensitivity of dew duration to meteorological parameters on a nightly time scale

The relationship between dew duration and RH (Fig. 7, panel a), $T_d - T_s$ (Fig 7, panel b) or wind speed (Fig. 8) is tested by assessing all data from the network of sites on a nightly time scale. These meteorological parameters have been averaged over the dew duration for each dewy night and their optimum values associated with long dew durations (6-15 h) will be calculated using the $50\,\%\{25\,\%, 75\,\%\}$ quantiles notation. We do not consider dew events longer than $15\,\text{h}$, because we expect that direct
solar radiation rapidly cancels dew formation. For example, LAJA has a maximum darkness duration of $\sim 14\,\text{h}$ on the study period (based on shortwave measurements).

For both ecosystems, $98\,\%$ of the dew events occur with a relative humidity above $60\,\%$ (2049 dewy nights). The maximum dew duration during the night is limited by the relative humidity and contained in a triangular envelope (Fig. 7, panel a), as it
has been shown using RDC sensors (Muselli et al., 2009; Lekouch et al., 2012; Maestre-Valero et al., 2015; Beysens, 2016). However, in these studies, the relative humidity was averaged over the night instead of the dew duration. The higher time resolution of our data and the diversity of the grassland and forested sites offer the opportunity to calculate the optimum value of RH to form dew. For long dew durations (between 6 and 15 h), the median relative humidity is $96\{93, 98\}\,\%$. It is remarkable to see that saturated air (RH $= 100\,\%$) is not beneficial to dew formation because of the cut-off of the radiative loss as previously
mentioned. We also report the relationship between dew duration and $T_d - T_s$ (Fig. 7, panel b) as previous studies have not had access to direct surface temperature measurements. A positive relationship can be observed when $T_d - T_s < 1\,°\text{C}$, meaning that longer dew events are associated with larger dew fluxes, which explains the quadratic pattern observed in Fig. 5 (panel b). However, the dew duration does not depend on $T_d - T_s$ anymore when $T_d - T_s > 1\,°\text{C}$ because it is now only limited by the maximum darkness duration (timing of sunrise and sunset, depending on the latitude and the seasons). A typical value of
$T_d - T_s$ for dew durations between 6 and 15 h is $1.3\{0.7, 2.0\}\,°\text{C}$. The relationship between dew yield and $T_d - T_a$ (instead of $T_d - T_s$) is drastically different because it shows a pattern with a triangular envelope (see Muselli et al., 2009; Lekouch et al., 2012; Maestre-Valero et al., 2015), highlighting the fact that air temperature is a poor proxy of the surface temperature.

Dew is also sensitive to the wind speed, and $99\,\%$ of the dew events occur while the wind speed is below $4\,\text{m}\,\text{s}^{-1}$ (1918
dewy nights). It is important to specify the canopy and measurement heights because the vertical logarithmic profile of the wind speed is affected by the roughness length. For the grasslands, the mean canopy height is $0.9 \pm 0.5\,\text{m}$ (1 SD) and the mean difference between the wind speed sensor height and the top of the canopy is $0.9 \pm 0.6\,\text{m}$ (1 SD). For the forests, as we are only interested in the dew events occurring within canopy, we have averaged the wind speed measurements from all sensors below the top of the canopy (see Fig. 1) and considered this spatial average as representative of the wind speed within canopy.
The next step is to compare the relationship between dew duration and wind speed for the two following groups: within the canopy of the forests (Fig. 8, panel a, 990 dewy nights) and at the top of the canopy of the grasslands (Fig. 8, panel b, 928 dewy nights). Figure 8 (panel c) shows graphically that the two groups exhibit a consistent triangular pattern despite various wind speed ranges, which indicates that they likely belong to the same population (Fig 8, panel d). As previously explained, a





minimum wind speed is required to sustain dew formation by replenishing water vapor molecules in the leaf boundary layer (first part of the triangle). On the other hand, strong wind speeds will equilibrate the temperature between the air and the surface at night and rapidly cancel dew formation (second part of the triangle). An optimum wind speed value is $0.5\{0.4, 0.9\}\,\mathrm{m\,s^{-1}}$ for dew durations between 6 and 15 h. The triangular envelope has been previously reported by Muselli et al. (2009); Lekouch et al. (2012) and Maestre-Valero et al. (2015) using the dew yield instead of the dew duration. Peaks in the dew yield occur

at similar wind speed when the wind sensor is at 2 m ($\sim 0.5\,\mathrm{m\,s^{-1}}$ in Maestre-Valero et al. (2015)). However, when the wind speed is measured at 10 m (Muselli et al., 2009) or extrapolated to 10 m (Lekouch et al., 2012), the peak is closer to $\sim 2\,\mathrm{m\,s^{-1}}$, which is comparable with our data assuming a logarithmic profile of wind speed.

### 3.3   Nocturnal cycles and timing of dew formation

We are now interested in the nocturnal cycles associated with the presence or absence of dew formation for three populations of grasslands (Fig. 9, dry, temperate and tropical) and two populations of forests (Fig. 10, early and late dew formation) from 16:00 to 10:00 next day. Three simple observations that confirm what has been noted in previous sections: (i) Dew rarely forms within the canopy of a grassland (Fig. 9, panels a.1 and b.1) and it rarely forms at the top of a canopy of a forest (Fig. 10, panels a.1 and b.1), (ii) The surface temperature, on average, does not increase during dew formation and (iii) The top and bottom of a

canopy of a forest have similar surface temperatures at night (the dashed and plain black lines are almost matching in Fig. 10). The first observation means that only the top of the canopy of a grassland cools at a sufficient rate for dew to form, and that the top of a canopy of a forest is too dry for dew to form. The second observation indicates that the radiative cooling (and possibly the ground flux acting sometimes as a sink of energy) is more important than the combined warming effects of condensation (associated with LE) and friction (associated with H) on a leaf during a dew event. The third observation means that forests

have a higher heat capacity as well as a higher water vapor content that redistributes radiative energy more efficiently and therefore makes the surface temperatures more homogeneous at night.

We focus in the following on the timing of dew formation. In the dry grassland sites (Fig. 9, panel a.1), the relative humidity is so low that the only time window for the dew point temperature to reach the surface temperature occurs at the end of the

night, when $T_s$ approaches its minimum (around 3:00-5:00). In temperate grassland sites (Fig. 9, panel b.1), the air is already warmer than the surface after 19:00 but it is not until midnight that the relative humidity rises above 85 % to start producing dew, which proceeds until 5:00. In tropical grassland sites (Fig. 9, panel c.1), dew (and presumably fog) starts around 22:00 and finishes at 7:00 ($\sim$ 2 h after sunrise). The population of forests with early dew formation (Fig. 10, panel a.1) is interesting, because the relative humidity can already reach 90 % at 21:00 and it allows dew to start forming. The downside of this high

level of humidity is a reduced rate of radiative cooling from 23:00 to 5:00 by 20 % compared to the other population shown in panel b.1 ($-44 \pm 3\,\mathrm{W\,m^{-2}}$ compared to $-54 \pm 3\,\mathrm{W\,m^{-2}}$). This stops dew formation at $\sim$2:00 because the surface temperature gets too warm compared to the air temperature. Finally, the population of forests (Fig. 10, panel b.1) will only produce dew



formation from midnight to 5:00.

The limitation of dew formation for the grasslands (Fig. 9, panels a.2 and b.2) is plain: the lower relative humidity yields a low dew point temperature and the higher wind speed enhances the sensible heat exchange so that the air and the surface approach equivalent temperatures. An analysis with similar results based on nocturnal cycles can be found in two studies (Meng and Wen, 2016; Zhuang and Zhao, 2017). However, the limitation of dew formation for the forests (Fig. 10) is less clear. The wind pattern within canopy is always the same ($\sim 0.5\,\mathrm{m\,s^{-1}}$) and the relative humidity from 23:00 to 5:00 is still high for panels a.2 and b.2 ($88\,\%$ and $86\,\%$). The major reason for the absence of dew formation in panels a.2 and b.2 is the low temperature difference between the air and the surface (within canopy), and this cannot be explained by the wind pattern, nor the radiative cooling. Indeed, from 23:00 to 5:00, the net radiation above the canopy is similar for panels a.1 vs a.2 ($-44\pm 3\,\mathrm{W\,m^{-2}}$ vs $-47\pm 2\,\mathrm{W\,m^{-2}}$, 1 SD) and slightly different for panels b.1 vs b.2 ($-54\pm 3\,\mathrm{W\,m^{-2}}$ vs $-49\pm 2\,\mathrm{W\,m^{-2}}$, 1 SD). Two important unknowns here are the cloud coverage, which will reduce the efficiency of the radiative cooling, and the ground flux, which will usually bring heat to the canopy. Further investigations on the relationship between surface temperature and air temperature in forested sites at night will be required to solve this problem.

## 3.4 Sensitivity of dew formation to the spatial variability of the surface temperature

Because the IR radiometer provides an average temperature for the surface of the canopy, we recognize that it may underestimate dew production because of the presence of cold points on the surface. Limited work that has been done using thermal imagery of canopies (Kim et al., 2018) has shown that during the day temperature can vary by multiple degrees based on the surface properties (e.g. the trunk, branches or dry soil versus wet soils or leaf surfaces). We do not have constraints on what the temperature distribution associated with each IR temperature measurement is because these spatial patterns depend on the architecture of the canopy, the wind speed, the stability of the atmosphere, the humidity and the plant type. Our assumption in this paper will be that the temperature range is distributed normally around the measured average, and we will test the sensitivity of dew frequency by varying the magnitude of the standard deviation of the normal distribution (from $\sigma = 0$, no spatial variability, to $\sigma = 0.5$, high spatial variability). As an example: a IR-radiometer measures $T_s = 14.0\,^{\circ}\mathrm{C}$ at 2:00 from the top of the canopy of a grassland, while the dew point temperature of the air is $T_d = 13.8\,^{\circ}\mathrm{C}$. Per definition, there is no dew formation ($T_d < T_s$) if we assume that there is no spatial variability in the surface temperature ($\sigma$=0), however $16\,\%$ of the canopy will receive dew formation for $\sigma = 0.2$ based on a Monte-Carlo Simulation (Fig. 11, panel a).

There is a linear response in the dew frequency for both the grasslands and forests to an increase in the standard deviation of the normal distribution representing the spatial variability of the surface temperature (Fig. 11, panels b and c). The mean percentage of the canopy receiving dew formation starts at $100\,\%$ for $\sigma = 0$ (all the canopy receives dew formation during a dew event when there is no spatial variability) and then decreases exponentially to 20-60$\,\%$ for $\sigma = 0.5$ depending on how wet the site is. We consider a $\sigma = 0.2$ for the grasslands ($95\,\%$ of the surface temperature anomalies within $[-0.4, 0.4]\,^{\circ}\mathrm{C}$, see panel





a) and $\sigma$=0.1 for the forests ($95\,\%$ of the surface temperature anomalies within $[-0.2, 0.2]\,°\mathrm{C}$ ), because it has been observed that the forests have more homogeneous surface temperatures at night than dry and temperate grasslands (Sect. 3.3). For these standard deviations, a typical percentage of the canopy receiving dew formation during a dew event is $64 \pm 7\,\%$ (1 SD) for the grasslands (tropical sites excluded) and $70 \pm 9\,\%$ (1 SD) for the forests. The dew frequency increases from $29 \pm 23\,\%$ (1 SD) to $35 \pm 26\,\%$ (1 SD) for the grasslands (tropical sites excluded) and from $25 \pm 14\,\%$ (1 SD) to $29 \pm 15\,\%$ (1 SD) for the forests. The

mean dew duration is minimally affected by considering the spatial variability of canopy temperatures, with a net gain of only $1\,\mathrm{h}$ with the highest variability ($\sigma = 0.5$) for the grasslands, and $1\,\mathrm{h}\,40$ for the forests. To the best of our knowledge, this sensitivity analysis of dew formation to the surface temperature variability has never been performed in the literature and it shows that the use of in-situ radiometric temperature seems robust to estimate the dew duration and dew frequency. Further analysis using thermal imaging would help to provide more realistic constraints in the distribution of temperature of the canopy at night.

## 4    Conclusions

This study is the first attempt at performing an analysis of dew formation in various ecosystems using IR radiometry from the National Ecological Observatory Network. The use of IR-radiometers in the study of dew formation was tested by Jacobs et al. (2006) for a single site, however our work emphasizes the importance of these continuous measurements for studying dew

on a continental scale over a long period of time. This method is more appropriate for ecosystem studies (as opposed to dew harvesting applications) because it does not rely on artificial surfaces and therefore provides measurements that capture the seasonal and annual dynamics of natural ecosystems. Moreover, the use of IR-radiometers allows us to bypass the technical difficulty of the surface temperature retrieval from an energy balance, which leads to a precise analysis of the sensitivity of dew formation to relative humidity and wind speed. It is always desirable to obtain direct measurements, however the difficulty

of obtaining large scale dew collection data makes this method very attractive. Further work would be then to cross check the present analysis with actual dew measurements.

The results obtained in our analysis are consistent with the previous dew studies, in terms of dew frequency, duration and yield. Additionally, the diversity of the environmental and climatic conditions for the 30 sites (Table 1 and Table 2) has led to

the discovery of new features that have not been reported in the literature yet. The main reason is that former studies tended to focus on areas where dew yield was expected to be high and because the lack of standardized measurements made it difficult to do quantitative cross-site syntheses. Four important results emerge from our analysis: (i) The dew frequency of grasslands follows a linear relationship with respect to the mean nocturnal RH (from 23:00 to 5:00) when RH $> 75\,\%$ (Fig. 4, panel a), (ii) Dew duration is maximum for values of RH $= 96\{93, 98\}\,\%$ and WS $= 0.5\{0.4, 0.9\}\,\mathrm{m\,s^{-1}}$ that do not seem to depend on the

ecosystem type (Fig. 7 and Fig. 8, $50\,\%\{25\,\%, 75\,\%\}$ quantiles notation) (iii) Dew duration and dew yield are related through a quadratic relationship (Fig. 5, panel b) and (iv) The spatial variability of the surface temperature does play an important role in the dew frequency, but has only a minimal affect on dew duration (Fig. 11). In parallel, one interesting result associated with





the nocturnal energy balance remains unresolved and will need further investigation: What is driving the exceptional difference between air and surface temperatures at night for certain sites even when the radiative cooling is inefficient (Fig. 3, panel c)? The answer will help to develop our understanding of energy and water exchange at night in natural ecosystems.

Some limitations are also present in this study. Firstly, the Monin-Obukhov Similarity Theory is not applicable to forested sites because the dewfall is occurring within the canopy and where estimating the aerodynamic resistance was not feasible. In additional, MOST produced unrealistically high estimates of yield for one tropical grassland site (DSNY). In these cases, it would be useful to use lysimeter data in order to validate the quadratic relationship between dew duration and dew yield using direct dew measurements (Fig. 5, panel b), and then parametrize the dew yield based on this relationship, using also the wind speed, canopy height and $T_d - T_s$. If this parametrization is sufficiently accurate to predict a dew yield, it will make for a simple approach to estimate dew yield that does not require modeling the resistance terms. The other limitation of our work is the absence of data such as cloud cover, precipitation and ground flux. The cloud cover would have been useful to explain the feature seen in $R_n$ vs RH (Fig. 3, panel a) or $T_a - T_s$ vs $R_n$ (Fig. 3, panel c) for some sites. The precipitation data are interesting because rainfall is an important source of moisture and it affects the surface temperature, and the ground flux is essential in closing the energy budget. These data will be updated by NEON in the future. Finally, some gaps of several days or weeks in the study period of most sites and the absence of frost events in our analysis did not allow us to perform a seasonal analysis. This result will be interesting in a future study, as dew formation has been shown to have a seasonal cycle (Zangvil, 1996; Xiao et al., 2009), driven by the $T_a$, RH and cloud coverage values, but also the maximum darkness duration.

Dew duration is very sensitive to the wind speed and relative humidity on site, and this raises the question on the impact of climate change and land surface use on the global dew yield. The intensification of the water cycle and the shift in the frequency of rainfalls are expected to drive changes in the surface relative humidity (Dai, 2006) and there has been a significant decrease observed in the surface wind speed in the Northern hemisphere over the past decades (Vautard et al., 2010). The global cloud coverage will also play a role in the efficiency of the nighttime radiative cooling, and its evolution remains difficult to constrain. In parallel, for a given relative humidity, grasslands are more likely to receive dew formation than forested sites because of a more efficient radiative cooling that depends on the canopy architecture and the specific heat. A future analysis taking these elements into account will be required to estimate the change of dew formation over the globe and to anticipate its impact on water stressed ecosystems that are endangered like the Californian Redwood forest or regions that are regreening like the Sahel (Dardel et al., 2014). Numerous studies have emphasized the ecological significance of dew formation, for example in the germination of seedlings (Zhuang and Zhao, 2016) or the recovery after a water-stress period (Munne-Bosch et al., 1999), and a network of continuous radiometric measurements such as the one built by NEON will be essential to produce high quality data for a global ecological study on non-rainfall water input.





*Code and data availability.* All data utilized in this study are freely available from the National Ecological Observatories Data repository (http://data.neonscience.org). Processed data products generated from these data and codes in R are available upon request to the author.

*Author contributions.* François Ritter designed the study, lead the data analysis, writing and figure generation. Max Berkelhammer assisted with study design, manuscript preparation and figure generation. Daniel Beysens assisted with data analysis and manuscript preparation.

*Competing interests.* The authors declare that they have no conflict of interest.

5  *Acknowledgements.* We would like to thank Josh Roberti and Morgan Jones for providing the meta-data of the sites. The work was partially supported through NSF grant 1502776 to Max Berkelhammer.





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





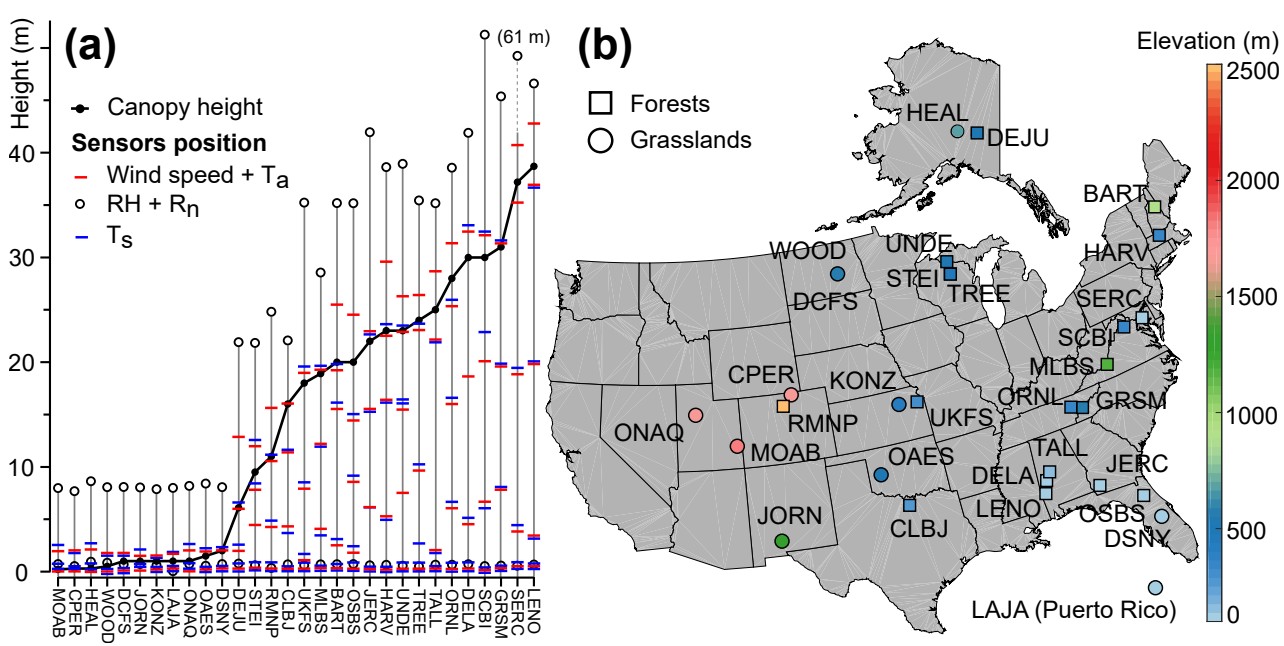

**Figure 1.** Panel a: vertical position of the sensors for each site (ordered with an increasing canopy height). Panel b: elevation and locality of the sites in the USA.





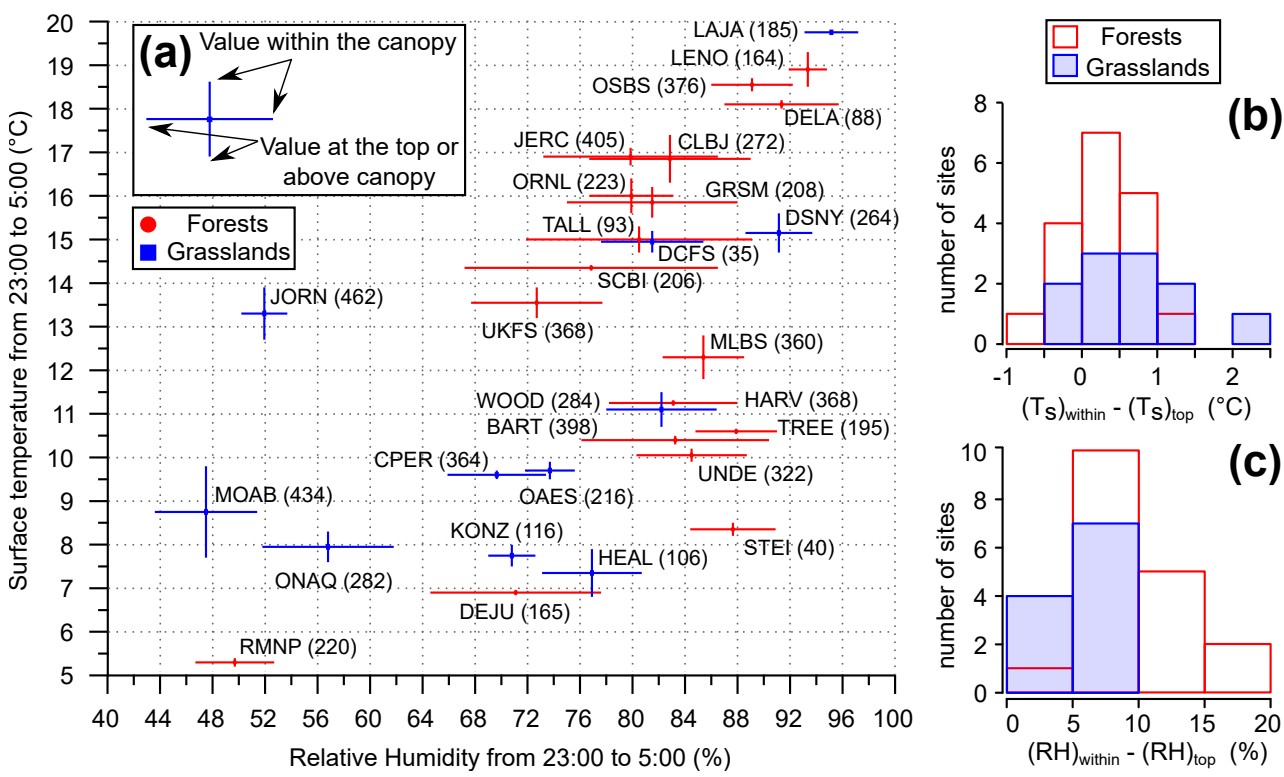

**Figure 2.** Panel a: $T_s$ versus RH from 23:00 to 5:00 next day over the growing season for each site (number of nights is indicated in parenthesis). Four mean values have been calculated per site, two from the sensors within canopy and two from the sensors at the top of canopy. Panels b (and c): histograms of the difference in $T_s$ (and RH) between the canopy within and the top of the canopy. SERC is missing because of a sensor failure above canopy.





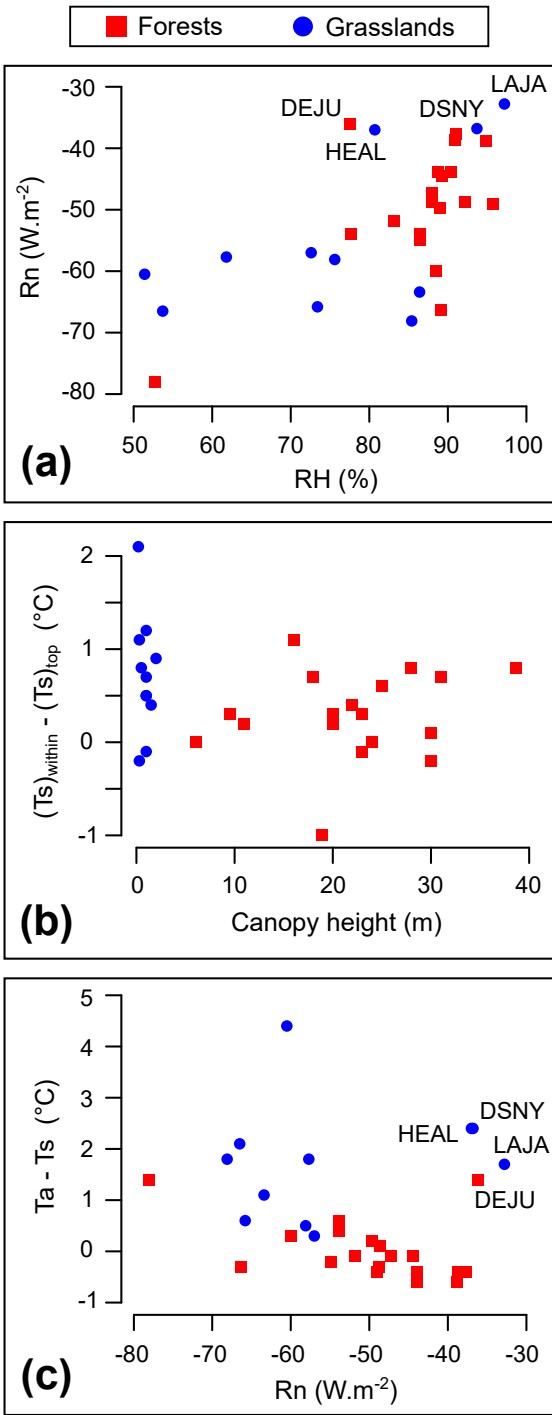

**Figure 3.** Mean values from 23:00 to 5:00 next day for each site over the growing season for the following parameters: $R_n$ versus RH (panel a), difference in $T_s$ between the canopy within and the top of the canopy versus the canopy height (panel b), $T_a - T_s$ (within canopy for the forests, top of the canopy for the grasslands) versus $R_n$ (panel c). Four sites are selected for discussion.

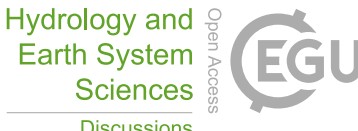



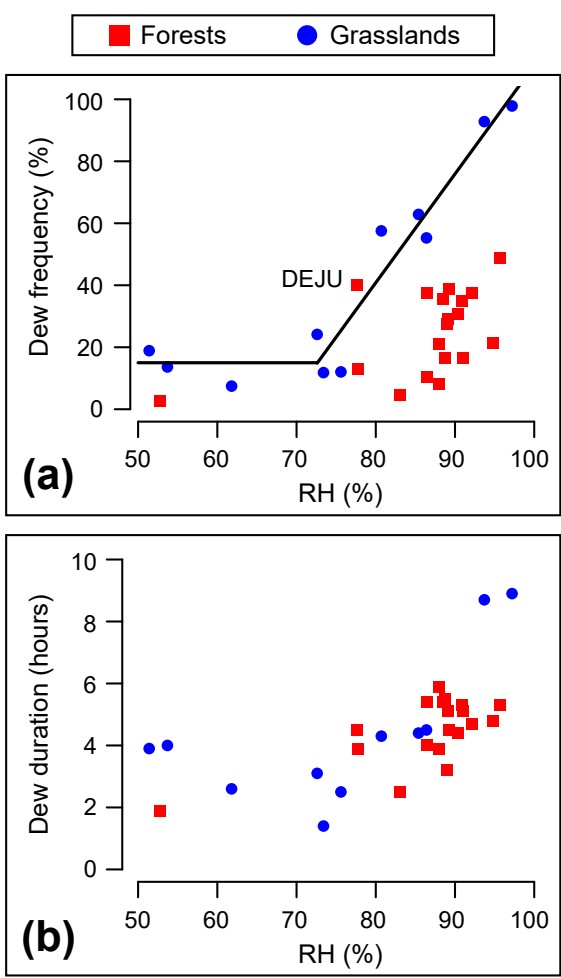

**Figure 4.** Dew frequency (panel a) and duration (panel b) versus RH (from 23:00 to 5:00 next day) for each site over the growing season. The slope of 3.5 ($r^2 = 0.92$) in panel a has been calculated based on the grasslands with RH > 70 % only. Dew formation has been calculated at the top of the canopy of the grasslands and within canopy of the forests only.



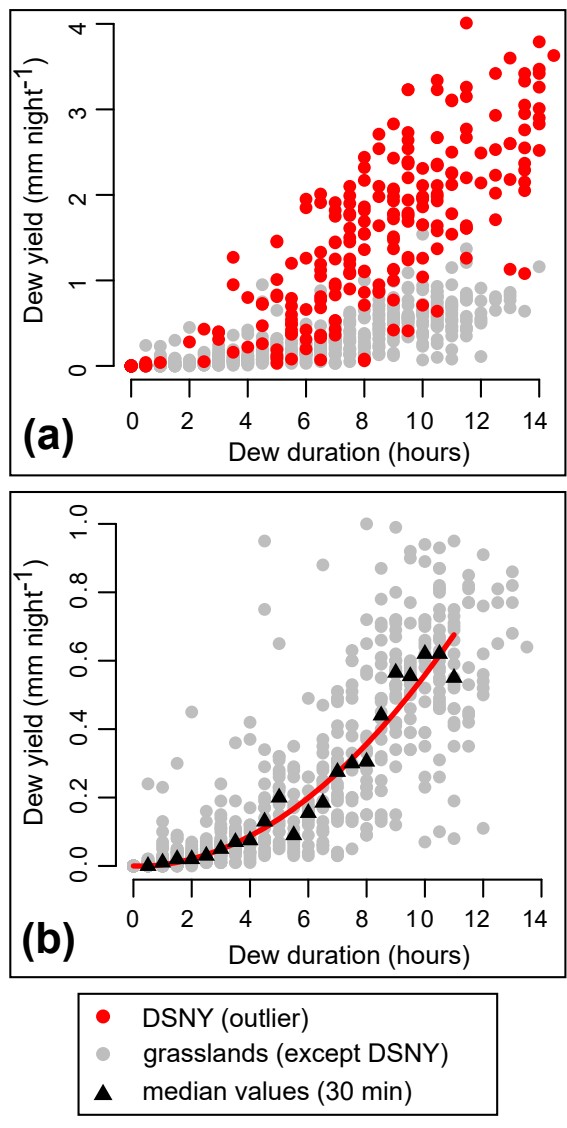

**Figure 5.** Dew yield versus dew duration for the grasslands (top of the canopy, nightly time scale). Panel a shows the failure of the Monin-Obukhov Similarity Theory for DSNY, that is why the quadratic fit ($y = a \times x^2, a = 0.0056 \, \mathrm{mm \, hours}^{-2}, r^2 = 0.94$) has been calculated for the other grasslands based on the 30-min median values for dew durations between 1 and 11 h (Panel b).





**Figure 6.** Median value (plain bars) and $\{25, 75\}\%$ quantiles (error bars) of the net radiation above canopy ($R_n$), the sensible (H) and latent heat fluxes (LE) averaged during a dew event for each dewy night (indicated in parenthesis). The sign convention is positive when the grassland receives energy and the residuals are $R_n + LE + H$.





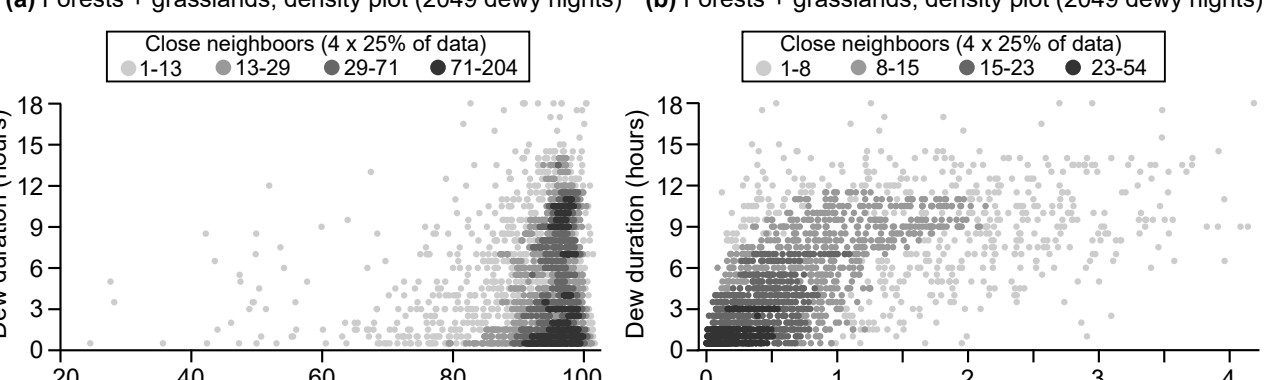

**Figure 7.** Density plot on a nightly time scale of the dew duration versus relative humidity (panel a) and dew duration versus $T_d - T_s$ (panel b), both averaged over the dew duration (all sites included). Four colors indicate the number of neighbors around a data point in a small rectangle ($\sim 2\%$ of the figure). Each color represents $25\%$ of the total number of nights.





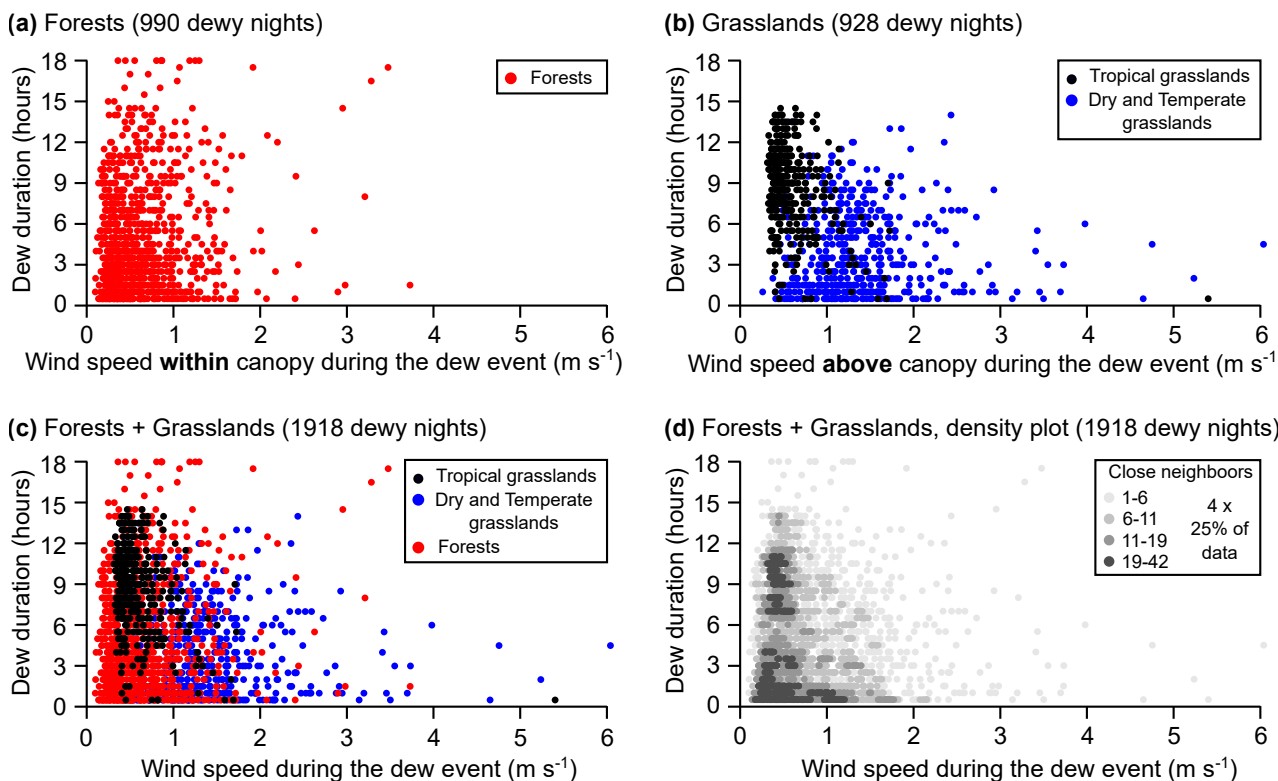

**Figure 8.** Dew duration versus wind speed averaged during a dew event for different groups (nightly time scale): forests (panel a, wind speed within canopy), grasslands (panel b, wind speed 1 m above canopy), both (panel c) and density plot of both (panel d). Density plot: four colors indicate the number of neighbors around a data point in a small rectangle ($\sim 2\%$ of the figure). Each color represents $25\%$ of the total number of nights.





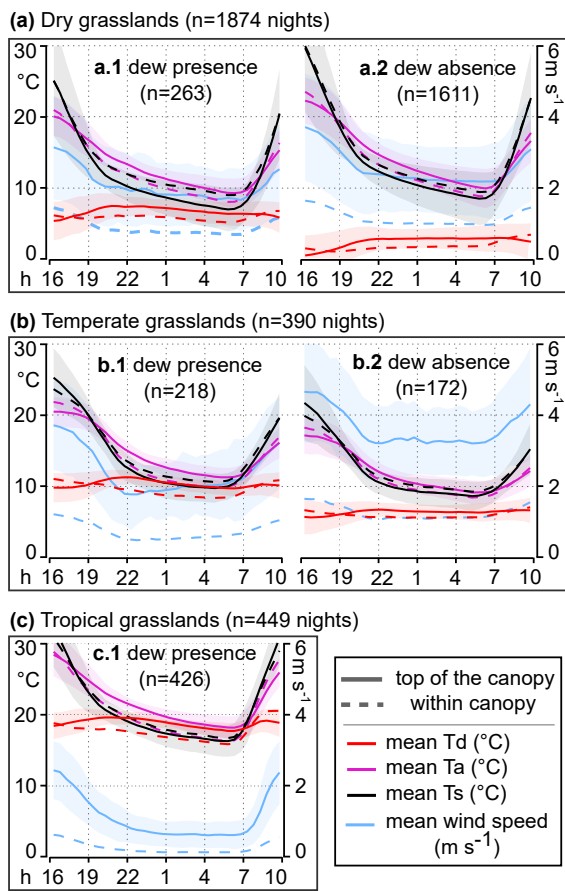

**Figure 9.** Nocturnal cycles for three populations of grasslands with the mean value of meteorological parameters calculated on a stack of n nights (n indicated in parenthesis). The plain lines represent the top of the canopy, the dashed lines within canopy. The shaded areas indicate $\pm 1$ standard deviation on the mean, shown for the top of the canopy only (for visualization purpose). The left column is based on nights with dew formation, the right column without. Panel c.2 is missing because of a too small sample size.





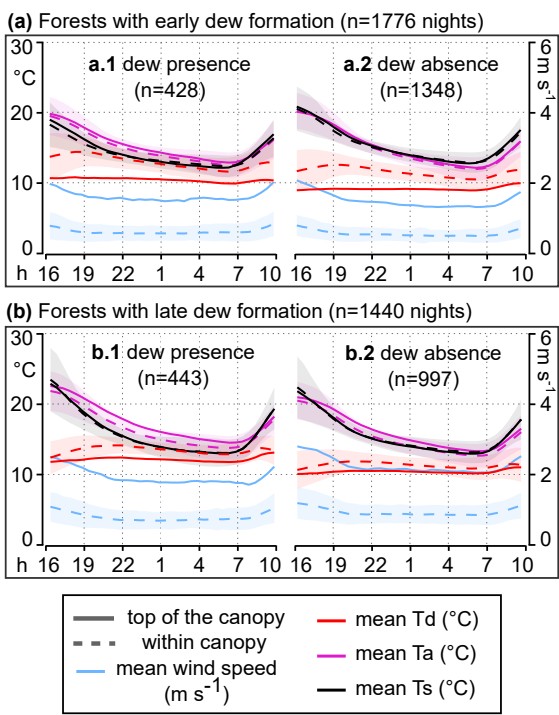

**Figure 10.** Nocturnal cycles for two populations of forests with the mean value of meteorological parameters calculated on a stack of n nights (n indicated in parenthesis). The plain lines represent the top of the canopy, the dashed lines within canopy. The shaded areas indicate $\pm 1$ standard deviation on the mean, shown within the canopy only (for visualization purpose). The left column is based on nights with dew formation, the right column without.





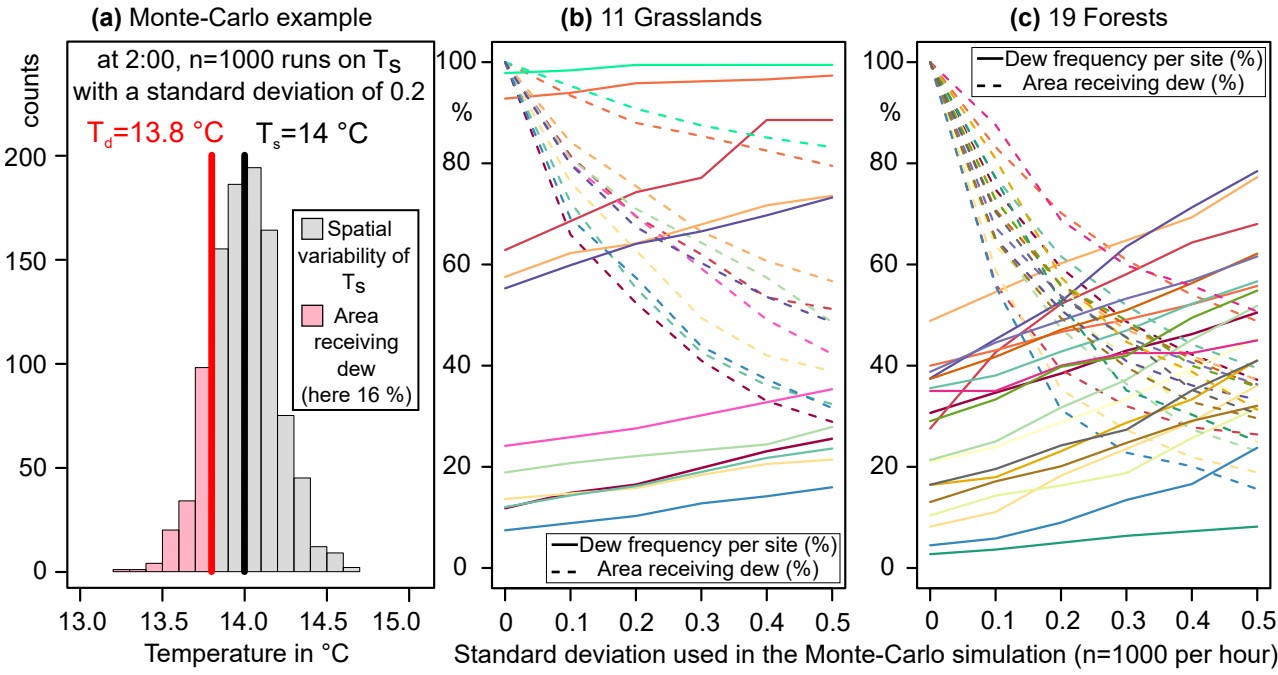

**Figure 11.** Sensitivity of dew frequency to the spatial variability in the surface temperature based on a Monte-Carlo simulation run on each half-hour data point (n=1000 runs). The spatial variability is assumed to follow a Normal distribution, and only a certain percentage of the canopy will receive dew formation (example for panel a, 16 % at 2:00). The Monte-Carlo simulation has been run with different standard deviations (from 0 to 0.5) for each half-hour for all grasslands (panel b) and all forests (Panel c). A non-dew event is defined here as less than 5 % of the canopy receiving dew formation, then all the areas above 5 % have been averaged for each site (dashed lines). Colors allow association between the plain lines and the dashed lines for each site.



**Table 1.** Summary of the results for the grassland sites ordered with an increasing RH. Dew frequency, duration, yield and meteorological parameters (averaged from 23:00 to 5:00 next day) have been calculated at the top of the canopy, except RH (close to the ground). C.h. stands for canopy height and elev. for elevation.

| sites | lat. | long. | elev. | nights | c.h. | RH | $T_a$ | $T_s$ | WS | $R_n$ | dew freq. | dew dur. | dew yield |
|---|---|---|---|---|---|---|---|---|---|---|---|---|---|
| | | | m | | m | % | °C | °C | $\mathrm{m\,s^{-1}}$ | $\mathrm{W\,m^{-2}}$ | % | h | $\mathrm{mm\,night^{-1}}$ |
| MOAB | 38.248 | -109.388 | 1799 | 434 | 0.2 | 51.4 | 12.1 | 7.7 | 2.2 | -60.5 | 19 | 3.9 | 0.10 |
| JORN | 32.591 | -106.843 | 1324 | 462 | 1.0 | 53.7 | 14.8 | 12.7 | 1.3 | -66.5 | 14 | 4.0 | 0.16 |
| ONAQ | 40.178 | -112.452 | 1663 | 282 | 1.0 | 61.8 | 9.4 | 7.6 | 1.8 | -57.7 | 7 | 2.6 | 0.04 |
| KONZ | 39.101 | -96.563 | 415 | 116 | 1.0 | 72.6 | 7.8 | 7.5 | 3.3 | -57.0 | 24 | 3.1 | 0.16 |
| CPER | 40.816 | -104.746 | 1654 | 364 | 0.3 | 73.4 | 10.3 | 9.7 | 2.7 | -65.8 | 12 | 1.4 | 0.01 |
| OAES | 35.411 | -99.059 | 519 | 216 | 1.5 | 75.6 | 10 | 9.5 | 2.9 | -58.1 | 12 | 2.5 | 0.14 |
| HEAL | 63.876 | -149.213 | 679 | 106 | 0.3 | 80.7 | 9.2 | 6.8 | 1.8 | -37 | 58 | 4.3 | 0.11 |
| DCFS | 47.162 | -99.107 | 576 | 35 | 1.0 | 85.4 | 16.5 | 14.7 | 2.5 | -68.1 | 63 | 4.4 | 0.42 |
| WOOD | 47.128 | -99.241 | 591 | 284 | 0.5 | 86.4 | 11.8 | 10.7 | 2.8 | -63.4 | 55 | 4.5 | 0.14 |
| DSNY | 28.125 | -81.436 | 20 | 264 | 2.0 | 93.7 | 17.1 | 14.7 | 0.8 | -36.8 | 93 | 8.7 | 1.45 |
| LAJA | 18.021 | -67.077 | 17 | 185 | 1.0 | 97.2 | 21.5 | 19.8 | 0.5 | -32.8 | 98 | 8.9 | 0.52 |





**Table 2.** Summary of the results for the forested sites ordered with an increasing RH. Dew frequency, duration and meteorological parameters (averaged from 23:00 to 5:00 next day) have been calculated within canopy, except $R_n$ (above canopy). C.h. stands for canopy height and elev. for elevation.

| sites | lat. | long. | elev. | nights | c.h. | RH | $T_a$ | $T_s$ | WS | $R_n$ | dew freq. | dew dur. |
|---|---|---|---|---|---|---|---|---|---|---|---|---|
| | | | m | | m | % | °C | °C | $m\,s^{-1}$ | $W\,m^{-2}$ | % | h |
| RMNP | 40.276 | -105.546 | 2742 | 220 | 11 | 52.7 | 6.8 | 5.4 | 1.2 | -78.1 | 3 | 1.9 |
| DEJU | 63.881 | -145.751 | 518 | 165 | 6 | 77.6 | 8.3 | 6.9 | 0.7 | -36.1 | 40 | 4.5 |
| UKFS | 39.04 | -95.192 | 322 | 368 | 18 | 77.7 | 14.5 | 13.9 | 1 | -53.9 | 13 | 3.9 |
| ORNL | 35.964 | -84.283 | 344 | 223 | 28 | 83.1 | 16.3 | 16.4 | 0.6 | -51.8 | 4 | 2.5 |
| JERC | 31.195 | -84.469 | 47 | 405 | 22 | 86.5 | 16.9 | 17.1 | 0.7 | -54.9 | 10 | 4.0 |
| SCBI | 38.893 | -78.139 | 353 | 206 | 30 | 86.5 | 14.8 | 14.4 | 1.1 | -53.9 | 37 | 5.4 |
| GRSM | 35.689 | -83.502 | 576 | 208 | 31 | 88 | 15.9 | 16.2 | 0.8 | -48.8 | 8 | 3.9 |
| HARV | 42.537 | -72.173 | 349 | 368 | 23 | 88 | 11.1 | 11.2 | 0.4 | -47.3 | 21 | 5.9 |
| MLBS | 37.378 | -80.525 | 1170 | 360 | 19 | 88.5 | 12.1 | 11.8 | 1.3 | -60 | 36 | 5.4 |
| UNDE | 46.234 | -89.537 | 522 | 322 | 23 | 88.7 | 9.8 | 10.2 | 0.7 | -43.9 | 16 | 5.5 |
| CLBJ | 33.401 | -97.57 | 273 | 272 | 16 | 89 | 17.6 | 17.4 | 0.6 | -49.7 | 28 | 3.2 |
| TALL | 32.95 | -87.393 | 167 | 93 | 25 | 89.1 | 15 | 15.3 | 1.5 | -66.3 | 29 | 5.1 |
| SERC | 38.89 | -76.56 | 33 | 276 | 37 | 89.3 | 13.2 | 13.3 | 0.5 | -44.5 | 39 | 4.5 |
| BART | 44.064 | -71.287 | 901 | 398 | 20 | 90.4 | 9.9 | 10.5 | 0.4 | -43.9 | 31 | 4.4 |
| STEI | 45.509 | -89.586 | 476 | 40 | 10 | 90.9 | 8.1 | 8.5 | 1.1 | -38.7 | 35 | 5.3 |
| TREE | 45.494 | -89.586 | 468 | 195 | 24 | 91 | 10.2 | 10.6 | 0.7 | -37.7 | 16 | 5.1 |
| OSBS | 29.689 | -81.993 | 46 | 376 | 20 | 92.2 | 18.8 | 18.7 | 0.4 | -48.7 | 38 | 4.7 |
| LENO | 31.854 | -88.161 | 15 | 164 | 39 | 94.8 | 18.7 | 19.3 | 0.4 | -38.8 | 21 | 4.8 |
| DELA | 32.542 | -87.804 | 25 | 88 | 30 | 95.7 | 17.6 | 18 | 0.7 | -49.0 | 49 | 5.3 |