# Peer review of "Dew frequency across the US from a network of in situ radiometers"

_Hydrology and Earth System Sciences, 2018_

## Referee Comment (RC1) · Anonymous Referee #1 · 4 Oct 2018

The manuscript "Dew frequency across the US from a network of in situ radiometers" by Ritter, Berkelhammer and Beysens describes an interesting analysis of data from a network of landsurface stations (NEON). The focus is on dew duration derived from radiative surface temperature and dew point at reference height. Also an attempt is made to come to quantitative estimates of dew yields. The topic is certainly relevant and also suitable for this journal.

General comments:

I am very enthusiastic about the set-up of the NEON network. One can hardly overestimate the importance of this type of networks where the same kind of measurements methods are applied in a standardized manner for various relevant eco-system types.

[Figure]

Regrettably there are no independent flux observations available to back up the dew yield estimates, which are now based on MO-theory. Although I acknowledge the difficulty of such observations, it would already help to verify the sensible heat flux estimate by 3D sonic anemometer/thermometer flux observations. I would welcome a disclaimer on this in the concluding section 4.

Specific comments

P4L12-13: The authors may want to refer to: Bosveld, F.C., A.A.M. Holtslag and B.J.J.M. van den Hurk, Night time convection in the interior of a dense Douglas-fir forest. Bound.-Layer Meteorol., 1999, 93, 171-195.

P4L16-18: Is the reverse also true. When no dew detected then the dew point temperature is below the radiometric surface temperature?

P5 Ch2. Methods: The IR-radiometers are essential for this study. Figure 1a gives a nice summary of instrumentation of the various sites used in this study. But it remains unclear what the IR-radiometers are actually "seeing". How are they directed what are their opening angles, please provide further information.

P5Ch2: Provide information on calibration (where how and how often).

P5L9: Discuss the accuracy of the RH sensors, especially in relation to the results shown in Figure 7. RH measurements during wet episodes are notoriously difficult.

P5L17: In the figure no RH sensor is shown at 15 m.

P5L18: "Canopy temperature" what is meant here, air or radiative temperature?

P5L29: The uncertainty of +_0.5oC in the Apogee sensor might as well by systematic between calibration events. Thus a simple sqrt(N) reduction of errors might be too optimistic, please comment.

P6L6: Due to the dew process a moisture gradient between RH observation level and the surface will occur. Please discuss the possible sensitivity of you duration estimates

in this light. P6L29 This tau is dimensionally incorrect, may be better to split this calculation 30x60 sec and then remark that the outcome is multiplied by 100 to arrive at mm.

Fig7 and 8: Please explain in the main text the method of "close neighbours".

P12L11-20: if I understand it correctly the authors estimate H and LE from MO-theory and Rn is measured. And then the residual is assigned to S-G. Please be more specific about this in the text P12L11-20.

P14 L13: But I see now drop of Ts below Td at all in these two figures?

P14L15: This might be related to downward convection from the crown level. See the same reference as given above (Bosveld etal, 1999).

---

## Referee Comment (RC2) · Anonymous Referee #2 · 5 Oct 2018

Ritter et al. presents new results extending the work of Jacobs et al. 2006 using IR radiometers to provide surface temperature and infer dew deposition. The authors use data from NEON sites to calculate dew deposition frequency and duration and infer characteristics of optimal dew deposition conditions. The analysis is sound and the results very interesting. The paper is well written, and I do not have any minor comments.

There is, however, one gaping hole in this paper: there is no actual dew deposition measurement presented against the model. I understand that the work presented here is based on Jacobs et al. 2006, in which the model is compared to real data, but I think the method and the instruments are uncommon enough that a second set of dew deposition data could be helpful. This could be easily achieved by temporarily placing

a leaf wetness sensor in the field at one of the NEON sites for example. The goal would be to verify that the predicted dew deposition times are correctly calculated by the model.

My second issue is with the lack of details regarding the model and its parameters. Right now, the paper refers to Jacobs et al. 2006, but since the model is really at the heart of the work presented here, I think it would be beneficial to flesh out the details of the different terms, either in the body of the paper or as an appendix. I would also like to see a more extended discussion on other types of dew formation models and how they get around the surface temperature issue. See for example Richards 2009, Maestro-Valero et al 2012, Gerlein-Safdi et al. 2018 that took different approaches to dew modeling.

Richards, K. (2009). Adaptation of a leaf wetness model to estimate dewfall amount on a roof surface. Agricultural and Forest Meteorology, 149(8), 1377–1383. http://doi.org/10.1016/j.agrformet.2009.02.014

Maestre-Valero, J. F., Ragab, R., Martínez-Alvarez, V., & Baille, A. (2012). Estimation of dew yield from radiative condensers by means of an energy balance model. Journal of Hydrology, 460-461(C), 103–109. http://doi.org/10.1016/j.jhydrol.2012.06.046

Gerlein-Safdi, C., Koohafkan, M. C., Chung, M., Rockwell, F. E., Thompson, S., & Caylor, K. K. (2018). Dew deposition suppresses transpiration and carbon uptake in leaves. Agricultural and Forest Meteorology, 259, 305–316. http://doi.org/10.1016/j.agrformet.2018.05.015

Section 3.1.2: Please include some significance test with your r^2 values. P11, L20: Why would a high vapor pressure deficit cause fog formation?

Results and Discussion: as is often the case when results and discussion are lumped together, the "discussion" part of the section ends up being short-sighted and too focused on the results, as opposed to promoting a reflection on the broader implications

of the work. What are the consequences of the results for ecosystem function and water balance? Some of this is shortly mentioned in the Conclusion but really belongs to a Discussion section. I would advise breaking down the Results and Discussion Sections and moving some of the points from the Conclusion into the Discussion, to make sure that the important takeaways from this work aren't lost in the middle of the (many) figure descriptions. . .

---

## Author Comment (AC1) · 15 Dec 2018

Dear anonymous referee 1,

First, thank you for your constructive comments and your interest in our study. We share the same disappointment concerning the absence of available flux measurements on the NEON sites, but this lack of data is temporary as NEON has planned to progressively update the database for the next years. This study is the first attempt of a continental dew analysis based on in-situ radiometric temperatures. It does not have the ambition to precisely constrain the nocturnal energy budget because too many parameters are unknown: density and emissivity of the canopy, cloud cover, ground

heat flux and independent latent and sensible flux estimations. Our conclusion now contains a stronger disclaimer on the goal and limitations of the study.

We have corrected all the minor comments you mentioned: The Bosveld et al. (1999) paper is now referenced, information on the IR-radiometers have been updated (angle, positioning), estimated uncertainties for the different measurements (RH included) have been added and a clarification has been brought to the close neighbors graph and the Fig. 6, the Energy balance. A couple of your comments required more attention:

"P5L29: The uncertainty of +/*0.5C in the Apogee sensor might as well by systematic between calibration events. Thus a simple sqrt(N) reduction of errors might be too optimistic, please comment. P6L6: Due to the dew process a moisture gradient between RH observation level and the surface will occur. Please discuss the possible sensitivity of you duration estimates."

- In this study, the possible statistical bias introduced by either the uncertainty on the radiometric surface temperature or the difference in RH between the sensor location and the boundary layer of the leaf is analyzed Section 3.4. The Monte-Carlo simulation allows to see how the frequency of the dew events is shifted for a given dispersion around the mean surface temperature value. This shift is estimated to be 5% for a sigma of 0.2 Celcius.

- Our dew yield estimation based on the MO-theory is identical to previous direct dew measurements: 0.14+/-0.12 mm/night versus 0.13+/-0.10 mm/night (Tomaszkiewicz et al., 2015). This gives a good confidence in our statistics.

Please let us know if you have additional comments,

Best regards,

François Ritter

Bosveld, F. C., Holtslag, A., and Van Den Hurk, B.: Nighttime convection in the interior of a dense Douglas fir forest, Boundary-Layer Meteorology, 93, 171–195, 1999.

———————————————

---

## Author Comment (AC2) · 15 Dec 2018

Dear anonymous referee 2,

Thank you for reviewing our study and for raising legitimate critics based on the lack of experimental validation of our continental dew analysis. We would like to oppose several answers to this:

- As explained in the introduction, dew formation is not a homogeneous process that can be easily measured with either radiative passive condensers or leaf wetness sensors. The emissivity, the size and the structure of these surfaces are different from

the canopy and they are not seasonally and annually evolving like the natural canopy. The only rigorous experimental validation of the dew yield in natural ecosystems is performed with lysimeters, which are expansive and provide local measurements. Jacobs et al. (2006) has validated the use of radiometric surface temperature to estimate dew formation with a direct comparison with micro lysimeters data.

- Our statistic is robust compared to previous work: the mean dew yield estimated with the Monin-Obukhov Theory in our study is 0.14+/-0.12 mm/night versus 0.13+/-0.10 mm/night from 25 previous dew studies (Tomaszkiewicz et al., 2015). Additionally, these dew yield estimations contain much more uncertainty than the dew frequency estimation because the leaf boundary layer needs to be modelized. Dew frequencies only require the dew point temperature of the air and the surface temperature, we therefore have a stronger confidence in these estimations.

- In anticipation of your comment, we have performed a Monte-Carlo simulation to quantify the sensitivity of dew frequency to a variability in the surface temperature. The result of the analysis shows a possible shift of 5% in the dew frequency for a sigma of 0.2 Celcius (see Sect. 3.4).

Based on these three points, we think that the term "gaping hole" you used in your review is in this context unjustified.

However, we do agree that the Method section was lacking details about Monin-Obukhov Similarity Theory. It has been updated with the explicit formula of the Bulk Richardson Number and the assumptions on the roughness lengths for vapor and momentum.

The dew formation models proposed by Richards 2009, Maestro-Valero et al 2012 or Gerlein-Safdi et al. 2018 are all based on an Energy balance, and they are now mentioned in the introduction.

The conclusion has also been modified. Now, a new section is present in the article that contains the discussion on the ecological significance of dew formation.

Please let us know if you have additional comments,

Best regards,

François Ritter

Jacobs, A. F. G., Heusinkveld, B. G., Kruit, R. J. W., and Berkowicz, S. M.: Contribution of dew to the water budget of a grassland area in the Netherlands, Water Resources Research, 42, 1–8, https://doi.org/10.1029/2005WR004055, 2006.

Tomaszkiewicz, M., Abou Najm, M., Beysens, D., Alameddine, I., and El-Fadel, M.: Dew as a sustainable non-conventional water resource: a critical review, Environmental Reviews, 23, 425–442, https://doi.org/10.1139/er-2015-0035, http://www.nrcresearchpress.com/doi/10.1139/er-2015-0035, 2015.

Richards, K. (2009). Adaptation of a leaf wetness model to estimate dewfall amount on a roof surface. Agricultural and Forest Meteorology, 149(8), 1377–1383. http://doi.org/10.1016/j.agrformet.2009.02.014

Maestre-Valero, J. F., Ragab, R., Martínez-Alvarez, V., Baille, A. (2012). Estimation of dew yield from radiative condensers by means of an energy balance model. Journal of Hydrology, 460-461(C), 103–109. http://doi.org/10.1016/j.jhydrol.2012.06.046

Gerlein-Safdi, C., Koohafkan, M. C., Chung, M., Rockwell, F. E., Thompson, S., Caylor, K. K. (2018). Dew deposition suppresses transpiration and carbon uptake in leaves. Agricultural and Forest Meteorology, 259, 305–316. http://doi.org/10.1016/j.agrformet.2018.05.015

---

## Author Response (AR1)

Dear Anonymous reviewers and dear Editor,

The general comments have previously been answered in the discussion, please find in the following our answer to the specific comments:

**For Reviewer #1:**

I would welcome a disclaimer on this [*independent flux observations*] in the concluding section 4.

Updated: *Direct flux observations are absent from this study and further work will be necessary to cross check the present analysis with actual dew measurements.*

P4L12-13: The authors may want to refer to: Bosveld, F.C., A.A.M. Holtslag and B.J.J.M. van den Hurk, Night time convection in the interior of a dense Douglas-fir forest. Bound.-Layer Meteorol., 1999, 93, 171-195.

Done.

P4L16-18: Is the reverse also true. When no dew detected then the dew point temperature is below the radiometric surface temperature?

Yes it is. We have modified the sentence to remove the ambiguity: "Two studies have validated the use of in-situ IR-radiometers and dew point temperature to predict the absence or presence of dew formation, by comparison with RDC and leaf wetness sensor data (Maestre-Valero et al., 2015) or lysimeter data (Jacobs et al., 2006)."

P5 Ch2. Methods: The IR-radiometers are essential for this study. Figure 1a gives a nice summary of instrumentation of the various sites used in this study. But it remains unclear what the IR-radiometers are actually "seeing". How are they directed what are their opening angles, please provide further information.

Their opening angle has been updated: "The height of the sensors depends on the site (see Ts in Fig. 1) and their pointing angle is usually 22 degrees from vertical."

P5Ch2: Provide information on calibration (where how and how often) + P5L9: Discuss the accuracy of the RH sensors, especially in relation to the results shown in Figure 7. RH measurements during wet episodes are notoriously difficult.

The calibration procedure is sophisticated and specifically detailed in the NEON documentation, so we were not able to develop this point in a short comment. However, NEON calculates an "expanded uncertainty" for each sensor that includes the calibration and the natural variability, and we have given the order of magnitude:

"For each data point, NEON provides an expanded uncertainty that includes calibration, data acquisition system, and natural variance. Typical nighttime values of these expanded uncertainties are +/- 0.6 C for Ts, +/- 2.2% for RH, +/- 0.1 m/s for WS, +/- 0.02 C for Ta and +/- 6 W/m2 for Rn."

P5L17: In the figure no RH sensor is shown at 15 m.

Updated.

P5L18: "Canopy temperature" what is meant here, air or radiative temperature?

Radiative temperature, it has been updated.

P5L29: The uncertainty of +_0.5oC in the Apogee sensor might as well by systematic between calibration events. Thus a simple sqrt(N) reduction of errors might be too optimistic, please comment.

This is partly why we have run a Monte Carlo simulation, to test the sensitivity of the predictions to a change in the surface temperature. A comment has been added.

P6L6: Due to the dew process a moisture gradient between RH observation level and the surface will occur. Please discuss the possible sensitivity of you duration estimates in this light.

Same comment than before, the Monte Carlo simulation will include this, because dew formation is predicted from Td-Ts, so if Ts varies with Td fixed, it will be similar to Td varying with Ts fixed. The sensitivity analysis will assess the impact of the possible difference in RH between the leaf boundary layer and the sensor boundary layer.

P6L29 This tau is dimensionally incorrect, may be better to split this calculation 30x60 sec and then remark that the outcome is multiplied by 100 to arrive at mm.

Updated.

Fig7 and 8: Please explain in the main text the method of "close neighbours".

Method updated with a new paragraph.

P12L11-20: if I understand it correctly the authors estimate H and LE from MO-theory and Rn is measured. And then the residual is assigned to S-G. Please be more specific about this in the text P12L11-20.

The Fig. 6 caption is now clearer about this point. Rn is detailed to be measured in the paragraph "description of the data" in the Method, and the calculation of H and LE through the MO-Theory is detailed in paragraph "Calculation of the dew yield, latent heat flux, sensible heat flux for the grasslands" in the Method.

P14 L13: But I see now drop of Ts below Td at all in these two figures?

Fig. 9 and 10 have been constructed by calculating the mean value of a stack of anomalies. If the mean value of Ts does not drop below the mean value of Td, it does not imply that it never happens on a nightly basis (that is why we show the standard deviation on the mean). There are some nights or hours in the night where Ts < Td but on average Ts > Td.

P14L15: This might be related to downward convection from the crown level. See the same reference as given above (Bosveld etal, 1999).

Very interesting comment, it has been updated: "Another explanation could be that a downward convection occurs from the crown level in stable conditions, therefore reducing the surface temperature difference via a sensible heat exchange (Bosveld et al., 1999)."

**For Reviewer #2:**

My second issue is with the lack of details regarding the model and its parameters. Right now, the paper refers to Jacobs et al. 2006, but since the model is really at the heart of the work presented here, I think it would be beneficial to flesh out the details of the different terms, either in the body of the paper or as an appendix.

The model and its parametrization are now much more detailed in the Method section (added: bulk Richardson number, parametrizations of the roughness lengths for vapor, momentum and heat). We did not develop the integrated stability functions for momentum and vapor as they belong to the classical Monin-Obukhov theory and they are well detailed in Jacobs et al. (2006).

I would also like to see a more extended discussion on other types of dew formation models and how they get around the surface temperature issue. See for example Richards 2009, Maestro-Valero et al 2012, Gerlein-Safdi et al. 2018 that took different approaches to dew modeling.

These three models are now included in the introduction. However, their approach is the same: an energy balance implicitly solved for $T_s$. It's simply the parametrization of the heat and vapor exchange that is changing, depending on the complexity of the model. In this study, we are lacking many information on the vegetation (density, architecture, plant type, seasonal variation ..) and we are focusing on a continental study, our model therefore needs to be simple and based on meteorological data. That is why we do not go into the details of dew modelling on a smaller scale because this is not the goal of this study.

Section 3.1.2: Please include some significance test with your rˆ2 values.

Done!

P11, L20: Why would a high vapor pressure deficit cause fog formation?

Thank you, it has been corrected.

Results and Discussion: as is often the case when results and discussion are lumped together, the "discussion" part of the section ends up being short-sighted and too focused on the results, as opposed to promoting a reflection on the broader implications of the work. What are the consequences of the results for ecosystem function and water balance? Some of this is shortly mentioned in the Conclusion

but really belongs to a Discussion section. I would advise breaking down the Results and Discussion Sections and moving some of the points from the Conclusion into the Discussion, to make sure that the important takeaways from this work aren't lost in the middle of the (many) figure descriptions

The conclusion and discussion have been reshaped and a new section "ecological significance of dew formation" has appeared in the discussion. Thank you for this comment.

**Compare Results**

| Old File: | | New File: |
|---|---|---|
| **OLD.pdf** | versus | **NEW.pdf** |
| **35 pages (1.76 MB)** | | **36 pages (1.77 MB)** |
| 9/5/18, 3:11:34 PM | | 1/11/19, 6:11:38 PM |

**Total Changes**

**316**

Text only comparison

**Content**

116   Replacements

103   Insertions

97   Deletions

**Styling and Annotations**

0   Styling

0   Annotations

**Go to First Change (page 1)**

[revised manuscript text omitted]
|---|---|---|---|---|---|---|---|---|---|---|---|---|
| RMNP | 40.276 | -105.546 | 2742 | 220 | 11 | 52.7 | 6.8 | 5.4 | 1.2 | -78.1 | 3 | 1.9 |
| DEJU | 63.881 | -145.751 | 518 | 165 | 6 | 77.6 | 8.3 | 6.9 | 0.7 | -36.1 | 40 | 4.5 |
| UKFS | 39.04 | -95.192 | 322 | 368 | 18 | 77.7 | 14.5 | 13.9 | 1 | -53.9 | 13 | 3.9 |
| ORNL | 35.964 | -84.283 | 344 | 223 | 28 | 83.1 | 16.3 | 16.4 | 0.6 | -51.8 | 4 | 2.5 |
| JERC | 31.195 | -84.469 | 47 | 405 | 22 | 86.5 | 16.9 | 17.1 | 0.7 | -54.9 | 10 | 4.0 |
| SCBI | 38.893 | -78.139 | 353 | 206 | 30 | 86.5 | 14.8 | 14.4 | 1.1 | -53.9 | 37 | 5.4 |
| GRSM | 35.689 | -83.502 | 576 | 208 | 31 | 88 | 15.9 | 16.2 | 0.8 | -48.8 | 8 | 3.9 |
| HARV | 42.537 | -72.173 | 349 | 368 | 23 | 88 | 11.1 | 11.2 | 0.4 | -47.3 | 21 | 5.9 |
| MLBS | 37.378 | -80.525 | 1170 | 360 | 19 | 88.5 | 12.1 | 11.8 | 1.3 | -60 | 36 | 5.4 |
| UNDE | 46.234 | -89.537 | 522 | 322 | 23 | 88.7 | 9.8 | 10.2 | 0.7 | -43.9 | 16 | 5.5 |
| CLBJ | 33.401 | -97.57 | 273 | 272 | 16 | 89 | 17.6 | 17.4 | 0.6 | -49.7 | 28 | 3.2 |
| TALL | 32.95 | -87.393 | 167 | 93 | 25 | 89.1 | 15 | 15.3 | 1.5 | -66.3 | 29 | 5.1 |
| SERC | 38.89 | -76.56 | 33 | 276 | 37 | 89.3 | 13.2 | 13.3 | 0.5 | -44.5 | 39 | 4.5 |
| BART | 44.064 | -71.287 | 901 | 398 | 20 | 90.4 | 9.9 | 10.5 | 0.4 | -43.9 | 31 | 4.4 |
| STEI | 45.509 | -89.586 | 476 | 40 | 10 | 90.9 | 8.1 | 8.5 | 1.1 | -38.7 | 35 | 5.3 |
| TREE | 45.494 | -89.586 | 468 | 195 | 24 | 91 | 10.2 | 10.6 | 0.7 | -37.7 | 16 | 5.1 |
| OSBS | 29.689 | -81.993 | 46 | 376 | 20 | 92.2 | 18.8 | 18.7 | 0.4 | -48.7 | 38 | 4.7 |
| LENO | 31.854 | -88.161 | 15 | 164 | 39 | 94.8 | 18.7 | 19.3 | 0.4 | -38.8 | 21 | 4.8 |
| DELA | 32.542 | -87.804 | 25 | 88 | 30 | 95.7 | 17.6 | 18 | 0.7 | -49.0 | 49 | 5.3 |

---

## Author Response (AR2)

Dear Anke Hildebrandt,

Thank you for your help in improving the quality of the manuscript.

I managed to contact a NEON engineer who explained to me that the expanded uncertainties are dynamically calculated and therefore the possible drift between two calibrations is considered. I have updated the method concerning this point.

I have added a disclaimer concerning the DSNY site saying that the failure of the Monin-Obukhov Similarity Theory might be related to the difficulty of measuring relative humidity in extremely moist conditions (DSNY is a tropical site). It was indeed a pertinent comment from the reviewer.

I have changed the name of the ecological section into "Ecological implications of the results".

The conclusion has been reworked, it does not contain abbreviations anymore, nor references to figures or tables.

Let me know if you are satisfied with this version,

Best regards,

Francois Ritter

[revised manuscript text omitted]